# Loss of TMEM106B and PGRN leads to severe lysosomal abnormalities and neurodegeneration in mice

Tuancheng Feng[1], Shuyi Mai[2], Jenn Marie Roscoe[1], Rory R Sheng[1], Mohammed Ullah[1], Junke Zhang[3], Isabel Iscol Katz[1], Haiyuan Yu[3], Wenjun Xiong[2] & Fenghua Hu[1,*] (iD)

## Abstract

Haploinsufficiency of progranulin (PGRN) is a leading cause of frontotemporal lobar degeneration (FTLD). Loss of PGRN leads to lysosome dysfunction during aging. *TMEM106B*, a gene encoding a lysosomal membrane protein, is the main risk factor for FTLD with PGRN haploinsufficiency. But how TMEM106B affects FTLD disease progression remains to be determined. Here, we report that TMEM106B deficiency in mice leads to accumulation of lysosome vacuoles at the distal end of the axon initial segment in motor neurons and the development of FTLD-related pathology during aging. Ablation of both PGRN and TMEM106B in mice results in severe neuronal loss and glial activation in the spinal cord, retina, and brain. Enlarged lysosomes are frequently found in both microglia and astrocytes. Loss of both PGRN and TMEM106B results in an increased accumulation of lysosomal vacuoles in the axon initial segment of motor neurons and enhances the manifestation of FTLD phenotypes with a much earlier onset. These results provide novel insights into the role of TMEM106B in the lysosome, in brain aging, and in FTLD pathogenesis.

**Keywords** frontotemporal lobar degeneration; lysosome; neurodegeneration; progranulin; TMEM106B

**Subject Categories** Molecular Biology of Disease; Neuroscience

See also: **X Zhou et al**, **G Werner et al** and **EL Clayton & AM Isaacs** (October 2020)

## Introduction

Haploinsufficiency of the progranulin (PGRN) protein, due to heterozygous mutations in the *granulin (GRN)* gene, is a leading cause of frontotemporal lobar degeneration with TDP-43 aggregates (FTLD-TDP) (Baker *et al*, 2006; Cruts *et al*, 2006; Gass *et al*, 2006). PGRN is known as a secreted glycoprotein of 7.5 granulin repeats (Bateman & Bennett, 2009; Cenik *et al*, 2012). Accumulating evidence has suggested a critical role of PGRN in the lysosome (Paushter *et al*, 2018). Firstly, patients with homozygous PGRN mutations exhibit neuronal ceroid lipofuscinosis (NCL), a lysosomal storage disorder (Smith *et al*, 2012; Almeida *et al*, 2016). Lipofuscin accumulation and lysosome abnormalities were also found in PGRN-deficient mice (Ahmed *et al*, 2010; Tanaka *et al*, 2014). Importantly, NCL-related phenotypes have been reported in FTLD patients with *GRN* mutation (Gotzl *et al*, 2014; Valdez *et al*, 2017; Ward *et al*, 2017), supporting that lysosomal dysfunction serves as a common mechanism for NCL and FTLD. Secondly, PGRN is a lysosome resident protein (Hu *et al*, 2010). PGRN interacts with another lysosomal protein prosaposin (PSAP) to facilitate each other's lysosomal trafficking (Zhou *et al*, 2015, 2017c). Upon reaching the lysosome, PGRN is processed to granulin peptides by cathepsins (Holler *et al*, 2017; Lee *et al*, 2017; Zhou *et al*, 2017b). PGRN and granulin peptides have been shown to regulate the activities of several lysosomal enzymes, including cathepsin D (Beel *et al*, 2017; Valdez *et al*, 2017; Zhou *et al*, 2017a; Butler *et al*, 2019) and glucocerebrosidase (Arrant *et al*, 2019; Valdez *et al*, 2019; Zhou *et al*, 2019). Thirdly, PGRN is transcriptionally co-regulated with a number of essential lysosomal genes by the transcriptional factor TFEB (Sardiello *et al*, 2009; Belcastro *et al*, 2011).

Frontotemporal lobar degeneration-TDP patients with *GRN* mutations show a high variability in age of onset and pathological presentation, even in cases with identical mutations (Van Deerlin *et al*, 2007), suggesting the existence of additional environmental or genetic factors that influence the disease manifestation. Genomewide association studies by several groups have identified *TMEM106B*, a gene encoding a type II transmembrane protein of unknown function, as a bona fide risk factor for FTLD, especially in patients with *GRN* mutations (Van Deerlin *et al*, 2010; Cruchaga *et al*, 2011; Finch *et al*, 2011; van der Zee *et al*, 2011). It has been proposed that TMEM106B functions as a risk factor for FTLD in a similar manner as APOE for Alzheimer's disease (Wood, 2010;

1   Department of Molecular Biology and Genetics, Weill Institute for Cell and Molecular Biology, Cornell University, Ithaca, NY, USA
2   Department of Biomedical Sciences, City University of Hong Kong, Kowloon, Hong Kong SAR, China
3   Department of Computational Biology, Weill Institute for Cell and Molecular Biology, Cornell University, Ithaca, NY, USA
   *Corresponding author. Tel: +86 607 2550667; Fax: +86 607 2555961; E-mail: fh87@cornell.edu

Cruchaga *et al*, 2011; van der Zee & Van Broeckhoven, 2011; Deming & Cruchaga, 2014; Jain & Chen-Plotkin, 2018). Cellular studies have revealed that TMEM106B is localized in late endosome/lysosome compartments and increased TMEM106B levels result in lysosome enlargement and dysfunction (Chen-Plotkin *et al*, 2012; Lang *et al*, 2012; Brady *et al*, 2013). These findings suggest that TMEM106B is critical for proper lysosomal functions, and as such, TMEM106B might modify FTLD phenotypes by regulating lysosomal functions.

Although *GRN* and *TMEM106B* were initially identified as genes associated with FTLD, these two genes have been intimately linked to brain health and associated with many other neurodegenerative diseases. First of all, *GRN* and *TMEM106B* have been discovered as the two main determinants of differential aging in the cerebral cortex with genome-wide significance (Rhinn & Abeliovich, 2017). Secondly, a recent study identified *GRN* and *TMEM106B* as two of the five risk factors for a recently recognized disease entity, limbic-predominant age-related TDP-43 encephalopathy (LATE) (Nelson *et al*, 2019). These two genes were also found to be associated with TDP-43 proteinopathy and coordinate with each other to regulate gene expression in a genome-wide transcription study (Yang *et al*, 2020). Furthermore, *GRN* and/or *TMEM106B* polymorphisms are associated with FTLD caused by *C9ORF72* mutations (van Blitterswijk *et al*, 2014; Deming & Cruchaga, 2014; Gallagher *et al*, 2014; Lattante *et al*, 2014), cognitive impairment in amyotrophic lateral sclerosis (ALS) (Vass *et al*, 2011) and Parkinson's disease (PD) (Baizabal-Carvallo & Jankovic, 2016; Tropea *et al*, 2019), and pathological presentation of Alzheimer's disease (AD) (Lee *et al*, 2011; Rutherford *et al*, 2012; Kamalainen *et al*, 2013; Perry *et al*, 2013; Sheng *et al*, 2014; Xu *et al*, 2017). Thus, it is critical to understand the functions of *GRN* and *TMEM106B* and how they genetically interact to affect neurodegeneration.

The TMEM106B risk allele has been reported to increase the levels of TMEM106B mRNA (Van Deerlin *et al*, 2010; Chen-Plotkin *et al*, 2012), suggesting that elevated TMEM106B levels increase the risk for FTLD-*GRN*. Consistent with this finding, in our previous studies we found that TMEM106B overexpression in neurons exacerbates lysosomal abnormalities caused by loss of PGRN (Zhou *et al*, 2017d). As such, lowering TMEM106B levels were thought to be beneficial to FTLD-*GRN* patients. This hypothesis was directly tested in a recent study, in which TMEM106B deficiency was shown to rescue lysosomal phenotypes, behavioral deficits, and retinal degeneration, but not lipofuscin accumulation and microglia activation associated with loss of PGRN in mouse models (Klein *et al*, 2017). However, another study has found that haploinsufficiency of TMEM106B cannot rescue phenotypes associated with PGRN haploinsufficiency (Arrant *et al*, 2018).

Here we demonstrate that mice deficient in both TMEM106B and PGRN show reduced motor activity, hindlimb weakness, and altered clasping behavior. Neuronal loss and severe microglia and astrocyte activation have been observed in the spinal cord, retina, and brain of these mice. Drastic autophagy and lysosomal abnormalities, and ALS/FTLD-related pathological changes have also been detected. Furthermore, TMEM106B deficiency alone leads to lysosome vacuolization in the axon initial segment (AIS) of motor neurons and ALS/FTLD-related phenotypes during aging. Taken together, these results provide novel insights into the role of PGRN and TMEM106B in brain aging and neurodegenerative diseases.

## Results

### Loss of TMEM106B results in lysosome defects in motor neurons and ALS/FTLD-related pathological changes

To determine the physiological function of TMEM106B, we generated a *Tmem106b*$^{-/-}$ mouse line using the CRISPR/Cas9 technique (Cong *et al*, 2013; Mali *et al*, 2013), in which a 341 bp fragment including the start codon was removed using two guide RNAs. Gene editing was verified by sequencing, PCR, and Western blot analysis using antibodies specific to TMEM106B (Feng *et al*, 2020). Loss of TMEM106B alone does not result in any obvious behavioral deficits in young mice, except slight motor coordination deficits on balance beams (Feng *et al*, 2020). A slight increase in GFAP levels, indicating astrocyte activation, as well as subtle changes in the levels of lysosomal proteins were also detected in TMEM106B-deficient mice (Feng *et al*, 2020).

To examine lysosome abnormalities in these mice more carefully, we stained brain sections and spinal sections with antibodies against lysosomal membrane protein LAMP1 and lysosomal protease cathepsin D. Enlarged LAMP1 and cathepsin D-positive vacuoles ranging from 2 to 22 μm were frequently observed in the ventral horn of spinal cord sections of 5-month-old TMEM106B-deficient mice. Immunostaining with the AIS marker neurofascin revealed that these vacuoles often accumulate at the distal end of the AIS in motor neurons (Fig 1A and B). These abnormal vacuoles were also observed in the facial motor nucleus, with increased diameters, consistent with the results from a recently published study (Luningschror *et al*, 2020). This suggests a critical role of TMEM106B in regulating lysosome dynamics specifically in motor neurons.

While no obvious changes in the levels of ubiquitinated proteins were detected in the 5-month-old TMEM106B-deficient mice, a significant increase in the levels of ubiquitinated proteins as well as autophagy adaptor protein p62 and phosphorylated TDP-43 (S403/S404) was observed in both brain and spinal cord lysates from the 16-month-old *Tmem106b*$^{-/-}$ mice (Fig 1C–F). These results suggest that TMEM106B deficiency leads to the development of ALS/FTLD-related pathology during aging.

---

**Figure 1. TMEM106B deficiency in mice results lysosome trafficking defects near AIS and ALS/FTLD-related pathology.**

A, B   Spinal cord sections from 5-month-old WT and *Tmem106b*$^{-/-}$ mice were stained with anti-neurofascin, anti- LAMP1 and anti-Cath D antibodies. Accumulation of LAMP1 and CathD-positive vacuoles at the distal end of AIS was observed in *Tmem106b*$^{-/-}$ motor neurons. Representative images from three independent mice were shown. Scale bar = 10 μm. *n* = 3. Data presented as mean ± SEM. Unpaired Student's *t*-test. ****$P < 0.0001$.

C–F   Western blot analysis of p62, ubiquitin (Ub), TDP-43, and p-TDP-43 in RIPA- and urea-soluble fractions from brain (C, E) and spinal cord (D, F) of 16-month-old WT and *Tmem106b*$^{-/-}$ mice. *n* = 5. Data presented as mean ± SEM. Unpaired Student's *t*-test. *$P < 0.05$, ***$P < 0.001$, ****$P < 0.0001$.

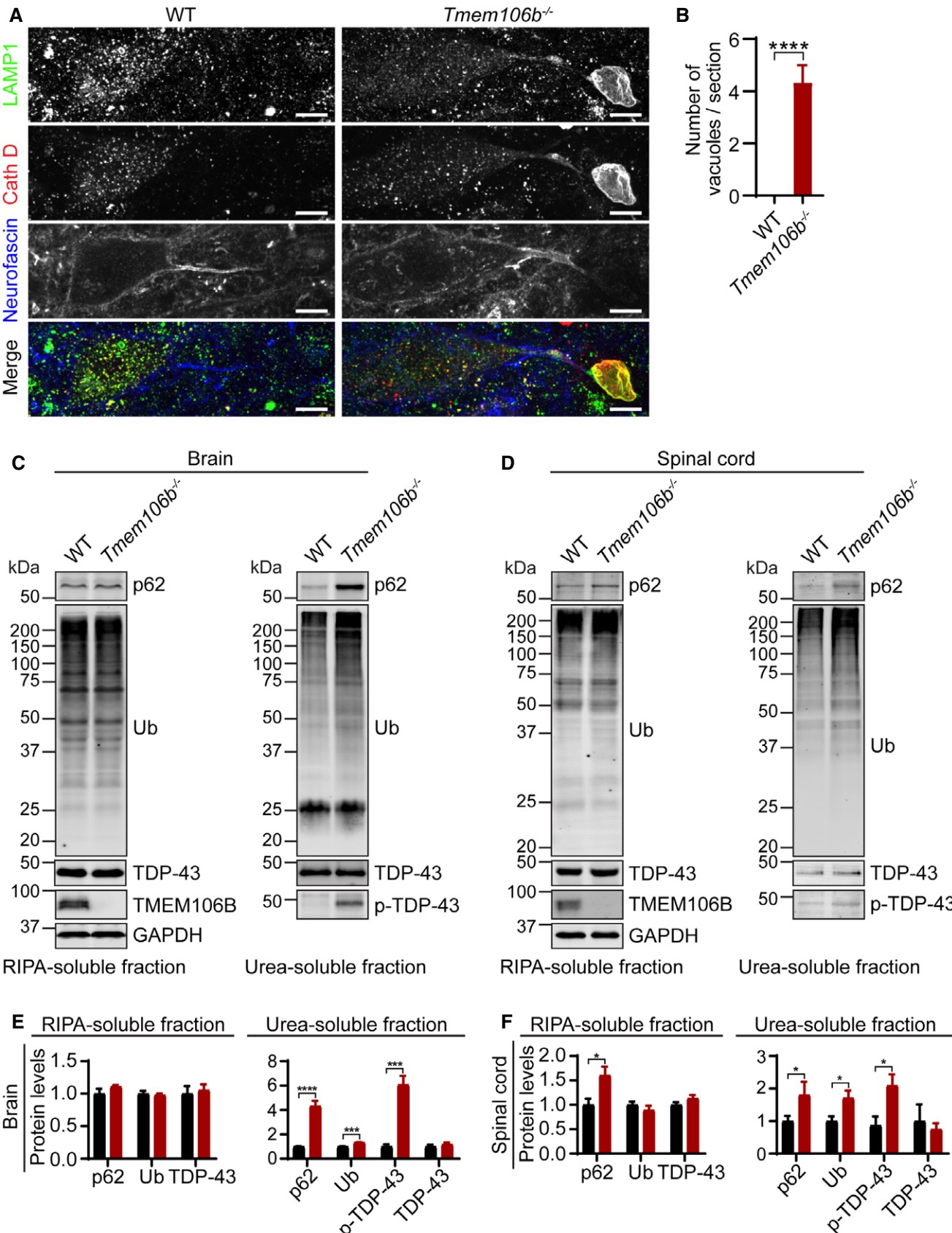

**Figure 1.**

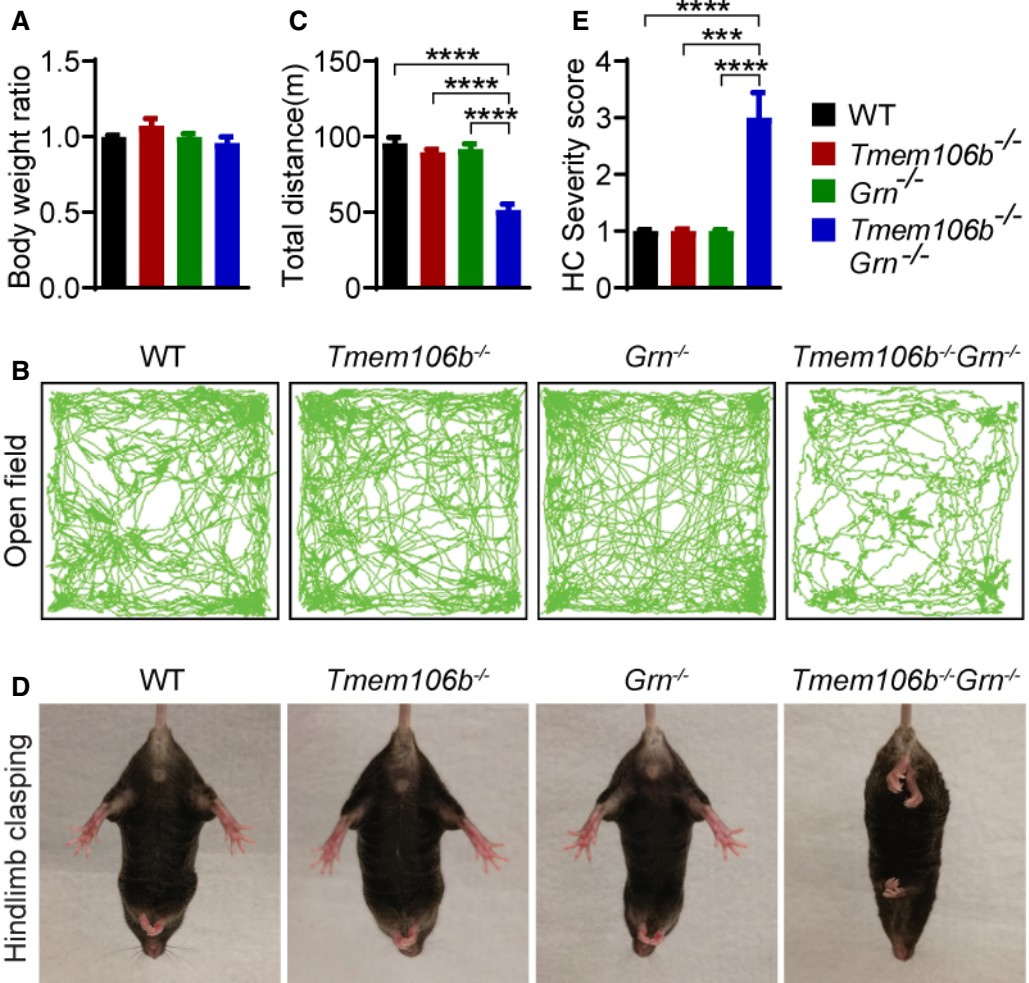

**Figure 2. Behavioral deficits of *Tmem106b*$^{-/-}$*Grn*$^{-/-}$ mice.**

A   *Tmem106b*$^{-/-}$*Grn*$^{-/-}$ mice do not show any changes in body weight. Body weight of 5-month-old mice of indicated genotypes were measured and analyzed. *n* = 6–10.

B, C   *Tmem106b*$^{-/-}$*Grn*$^{-/-}$ mice show reduced activities in the open-field test. 4.5-month-old mice were used in the study. Total movement was quantified. *n* = 8–10. Data presented as mean ± SEM. One-way ANOVA tests with Bonferroni's multiple comparisons: ****$P$ < 0.0001.

D, E   *Tmem106b*$^{-/-}$*Grn*$^{-/-}$ mice show abnormal hindlimb clasping behavior. 5-month-old mice were used in the study. Severity score was quantified. *n* = 8–10. Data presented as mean ± SEM. One-way ANOVA tests with Bonferroni's multiple comparisons: ***$P$ < 0.001, ****$P$ < 0.0001.

Data information: For all the behavioral studies, mixed male and female mice were used.

## Ablation of both PGRN and TMEM106B leads to severe motor defects, neurodegeneration and microglia and astrocyte activation

Next, we generated mice deficient in both PGRN and TMEM106B by crossing *Grn*$^{-/-}$ mice (Yin *et al*, 2010a) with the *Tmem106b*$^{-/-}$ mice that we had previously generated. *Tmem106b*$^{-/-}$*Grn*$^{-/-}$ mice are born at normal Mendelian frequency and do not show any obvious growth defects or body weight changes (Fig 2A). However, starting at around 3.5 months of age, these mice develop severe ataxia, hindlimb weakness, and motor defects (Movies EV1 and EV2). Motor activities were severely impaired as shown in the open-field test (Fig 2B and C). In addition, these mice have an abnormal hindlimb clasping behavior (Fig 2D and E), which is often observed

in mice with lesions in the spinal cord, cerebellum, basal ganglia, and neocortex (Lalonde & Strazielle, 2011). By 5.5 months of age, these mice have to be sacrificed. No obvious hindlimb weakness or ataxia phenotypes were observed in 5-month-old and 16-month-old *Tmem106b*$^{+/-}$*Grn*$^{-/-}$ and *Tmem106b*$^{-/-}$*Grn*$^{+/-}$ mice (our unpublished observations), suggesting that total loss of both PGRN and TMEM106B is required for the development of these behavioral phenotypes.

The strong behavioral deficits of *Tmem106b*$^{-/-}$*Grn*$^{-/-}$ mice indicate neuronal dysfunction. Immunostaining and Western blot analysis with antibodies against neuronal marker NeuN revealed a significant loss of neurons in the *Tmem106b*$^{-/-}$*Grn*$^{-/-}$ spinal cord, accompanied by severe microglia and astrocyte activation with increased intensities of IBA1 and GFAP, the microglia and astrocyte

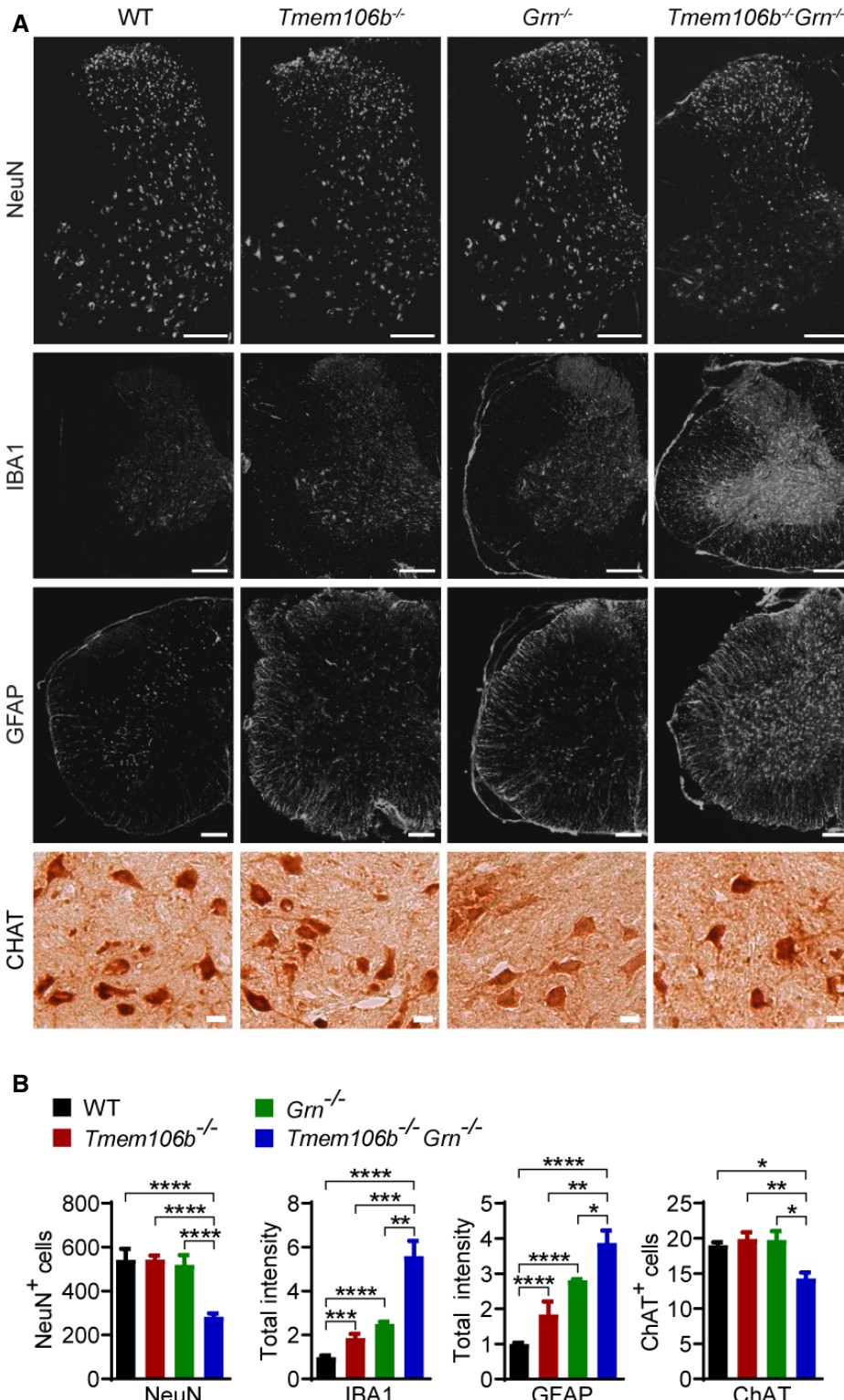

**Figure 3. Neuronal loss and increased gliosis in *Tmem106b*$^{-/-}$*Grn*$^{-/-}$ spinal cord.**

A, B Spinal cord sections (C1–C4) from 5-month-old mice of indicated genotypes were stained with indicated antibodies (A) and the numbers of NeuN, or CHAT-positive cells, as well as IBA1 and GFAP intensities, were quantified (B). Three–five sections/mouse were used for quantification. $n = 3$. Data presented as mean ± SEM. One-way ANOVA tests with Bonferroni's multiple comparisons: *$P < 0.05$, **$P < 0.01$, ***$P < 0.001$, ****$P < 0.0001$. Scale bar: 20 µm for CHAT staining, 200 µm for other images.

markers, respectively (Figs 3A and B, and 4A and B). Loss of motor neurons in the ventral horn region was visualized using antibodies against choline acetyltransferase (CHAT) (Fig 3A and B). A significant decrease in the protein levels of postsynaptic marker PSD95 and presynaptic marker synaptophysin (SYN) was also observed in the spinal cord lysates (Fig 4A and B), indicating the loss of synapses. These data suggest that the simultaneous loss of PGRN and TMEM106B leads to severe neurodegeneration and microglia and astrocyte activation in the spinal cord. Similar, but milder, phenotypes were observed in the brain with a reduction in NeuN and an increase in GFAP protein levels, but the levels of PSD95 and SYN were not affected (Fig 4C and D). In the cerebellum, there were no obvious changes in the numbers of Purkinje cells and cerebellar granule neurons (Fig EV1A and B), but significant microglia and astrocyte activation was observed, as indicated by an increase in GFAP and IBA1 staining intensities (Fig EV1C and D).

Progranulin has been shown to be neuroprotective in retinae (Tsuruma *et al*, 2014). PGRN deficiency leads to neuronal loss and lipofuscin accumulation in the retina in both humans and mice (Hafler *et al*, 2014; Ward *et al*, 2014, 2017). To determine whether TMEM106B ablation exacerbates these phenotypes, we examined the retinal sections from 5-month-old $Tmem106b^{-/-}Grn^{-/-}$ mice. Consistent with previous findings, PGRN ablation leads to the loss of photoreceptors, indicated by the thinning of the outer nuclear layer (ONL), which is mostly composed of nuclei of the rod photoreceptors, and this phenotype is enhanced in the $Tmem106b^{-/-}Grn^{-/-}$ mice (Fig 5A and B). We further stained retinal sections with antibodies against red/green and blue opsins to label all cone photoreceptor cells. Although the number of cone cells did not decrease drastically in the single- or double-knockout mice, a significant reduction in the length of the cone outer segment (COS) was observed in retinas from $Tmem106b^{-/-}Grn^{-/-}$ mice (Fig 5A and B), suggesting that cone cells are also affected in the double-knockout mice. Retinal ganglion cells, as indicated by Brn3a staining, do not seem to be affected in the $Grn^{-/-}$ or $Tmem106b^{-/-}Grn^{-/-}$ mice (Fig 5A and B). TUNEL staining revealed increased apoptosis in $Tmem106b^{-/-}Grn^{-/-}$ retinae (Fig 5C and D). Microglia activation and Müller cell gliosis—shown by IBA1 and GFAP staining, respectively—were observed in $Grn^{-/-}$ mice, but they were significantly increased in $Tmem106b^{-/-}Grn^{-/-}$ mice (Fig 5E–H). Taken together, these data support that the loss of TMEM106B exacerbates neuronal loss and gliosis phenotypes in the retinae of the $Grn^{-/-}$ mice.

## Increased expression of inflammation and lysosome genes in $Tmem106b^{-/-}Grn^{-/-}$ mice

To dissect molecular pathways affected by the loss of both TMEM106B and PGRN, we performed RNA-Seq analyses of cortex and spinal cord samples from 2.7-month-old WT, $Tmem106b^{-/-}$, $Grn^{-/-}$, and $Tmem106b^{-/-}Grn^{-/-}$ mice. Loss of TMEM106B or PGRN alone results in minimal changes in gene expression at this young age, but loss of both TMEM106B and PGRN leads to differential expression of inflammatory and lysosomal genes in the cortex (Table EV1 and Fig 6A), and spinal cord (Dataset EV1 and Fig 6B). Among the 12 genes upregulated in the $Tmem106b^{-/-}Grn^{-/-}$ cortex samples, inflammatory and lysosomal pathways are the two pathways mainly represented (Fig 6A). These two pathways are also significantly enriched in the spinal cord samples (Figs 6C and

EV2A), with many differentially expressed genes (DEGs) encoding cytokine and cytokine receptors, complement system proteins, immune signaling proteins, and lysosomal proteins. Among the 320 downregulated genes in the spinal cord, *Ap1s2*, which encodes the sigma-2 subunit of the heterotetrameric adaptor protein-1 (AP-1) complex, is the only lysosomal gene. Other downregulated genes include many genes involved in synapse formation and function (Fig EV2B), which might reflect changes in neuronal functions in these mice.

Many genes upregulated in $Tmem106b^{-/-}Grn^{-/-}$ samples belong to the gene signature associated with microglia activation during neurodegeneration, so-called disease-associated microglia (DAM) or neurodegenerative microglia (mGnD)(Keren-Shaul *et al*, 2017) (Krasemann *et al*, 2017; Deczkowska *et al*, 2018), including *Trem2, Tyrobp, Lgals3, Cd68,* and *Gpnmb* (Figs 6A and C and EV2C and D, Table EV2). The upregulation of the galectin-3 protein, encoded by the *Lgals3* gene, and GPNMB protein was confirmed by Western blot analysis of spinal cord and brain lysates. Both galectin-3 and GPNMB show a specific upregulation in the $Tmem106b^{-/-}Grn^{-/-}$ spinal cord and brain lysates, with minimal changes in the $Tmem106b^{-/-}$ and $Grn^{-/-}$ samples (Fig 7A–D), supporting that the loss of both PGRN and TMEM106B leads to excessive microglia activation. Immunostaining with antibodies against galectin-3, CD68, and cathepsin D (CathD) showed a significant overlap in the expression patterns of these proteins in the microglia, suggesting that these proteins are co-regulated in $Tmem106b^{-/-}Grn^{-/-}$ mice (Fig 7E).

## Lysosome abnormalities in $Tmem106b^{-/-}Grn^{-/-}$ mice

Our RNA-Seq analysis also showed increased expression of many lysosomal genes in the $Tmem106b^{-/-}Grn^{-/-}$ cortex (Fig 6A and Table EV1) and spinal cord samples (Figs 6C and EV2, and Dataset EV1), indicating that the lysosomal pathway is affected in these mice. Lipofuscin accumulation is an indication of lysosome dysfunction, and PGRN deficiency has been known to increase lipofuscin accumulation (Ahmed *et al*, 2010; Tanaka *et al*, 2014). An additional increase in lipofuscin signals was observed in the retina (Fig 8A and B), spinal cord (Fig 8C and D), and thalamus (Fig 8E and F) of $Tmem106b^{-/-}Grn^{-/-}$ mice compared to those of $Grn^{-/-}$ mice, indicating that TMEM106B ablation further exacerbates lysosome abnormalities in PGRN-deficient mice. Furthermore, a drastic upregulation of lysosomal proteases, cathepsins B, D, and L, was observed in spinal cord lysates (Fig 8G and H) and to a lesser extent, in the brain lysates (Fig EV3A and B) from $Tmem106b^{-/-}Grn^{-/-}$ mice. An upregulation of lysosomal membrane proteins, LAMP1 and LAMP2, was observed in spinal cord lysates but not the brain lysates (Figs 8G and H, and EV3A and B).

Exacerbation of lysosome abnormalities in $Tmem106b^{-/-}Grn^{-/-}$ mice suggests a non-overlapping role of PGRN and TMEM106B in regulating lysosome function. Since we have shown lysosome vacuolization in the AIS region of the $Tmem106b^{-/-}$ motor neurons (Fig 1A and B), we co-stained LAMP1, cathepsin D (CathD), and the AIS marker neurofascin in $Grn^{-/-}$ and $Tmem106b^{-/-}Grn^{-/-}$ spinal cord sections. An increased accumulation of LAMP1 and cathepsin D-positive vacuoles at the distal end of the AIS of motor neurons was observed in the $Tmem106b^{-/-}Grn^{-/-}$ spinal cord sections,

compared to those from age-matched $Tmem106b^{-/-}$ mice (Fig 9A and B). This phenotype was not observed in spinal cord sections from PGRN deficient mice (Fig 9A). These data indicate that loss of

both PGRN and TMEM106B exacerbates lysosome abnormalities in axons, which is likely to contribute to neuronal dysfunction and neuronal death in these mice.

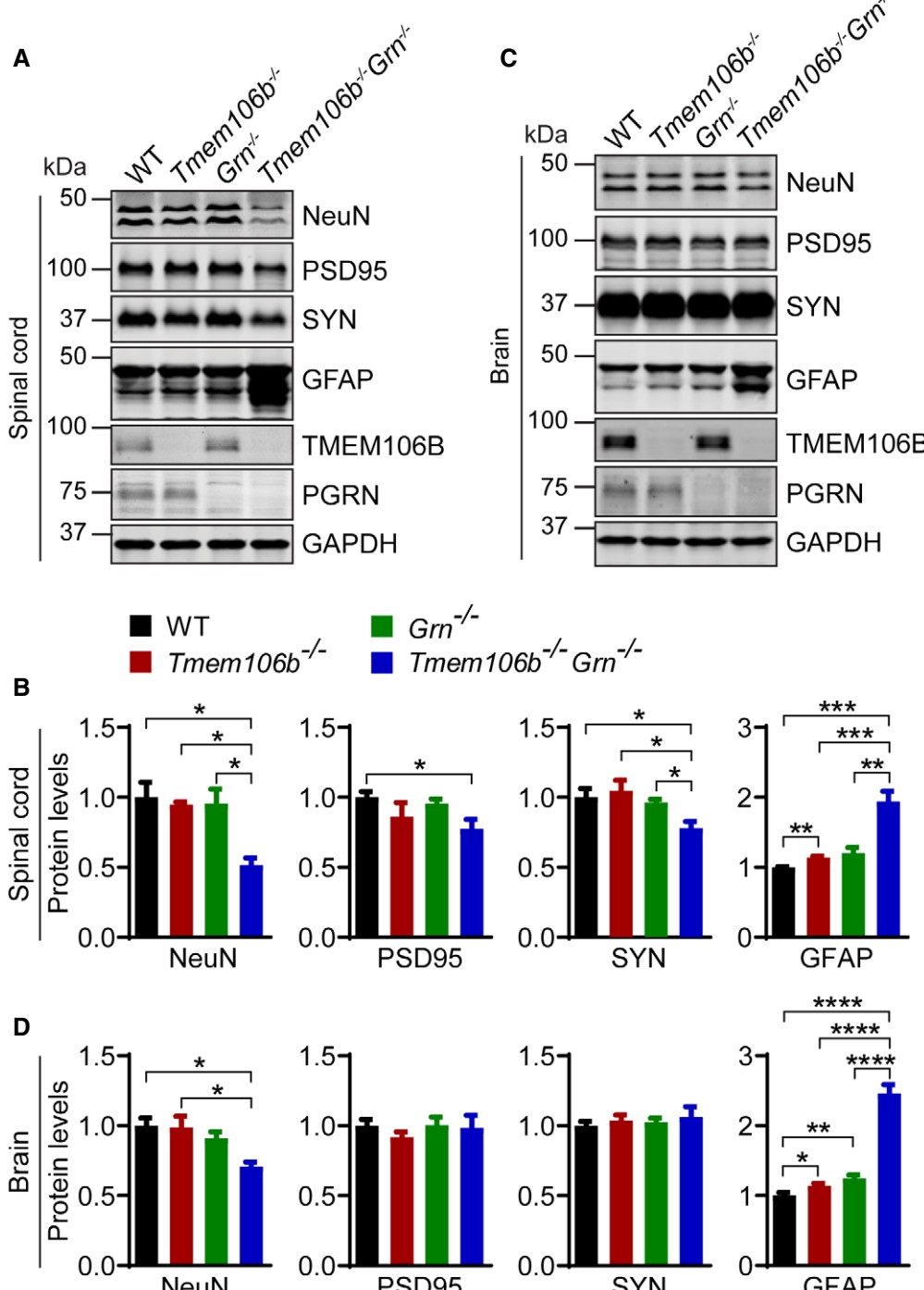

**Figure 4. Altered protein levels of NeuN, PSD95, SYN, and GFAP in $Tmem106b^{-/-}Grn^{-/-}$ mice.**

A–D   Western blot analysis of NeuN, PSD95, SYN, GFAP, TMEM106B, PGRN, and GAPDH in spinal cord (C5–C8) (A, B) and brain (C, D) lysates from 5-month-old WT, $Tmem106b^{-/-}$, $Grn^{-/-}$, and $Tmem106b^{-/-}Grn^{-/-}$ mice.

Data information: $n = 3$. Data presented as mean ± SEM. One-way ANOVA tests with Bonferroni's multiple comparisons: *$P < 0.05$, **$P < 0.01$, ***$P < 0.001$, ****$P < 0.0001$.

Source data are available online for this figure.

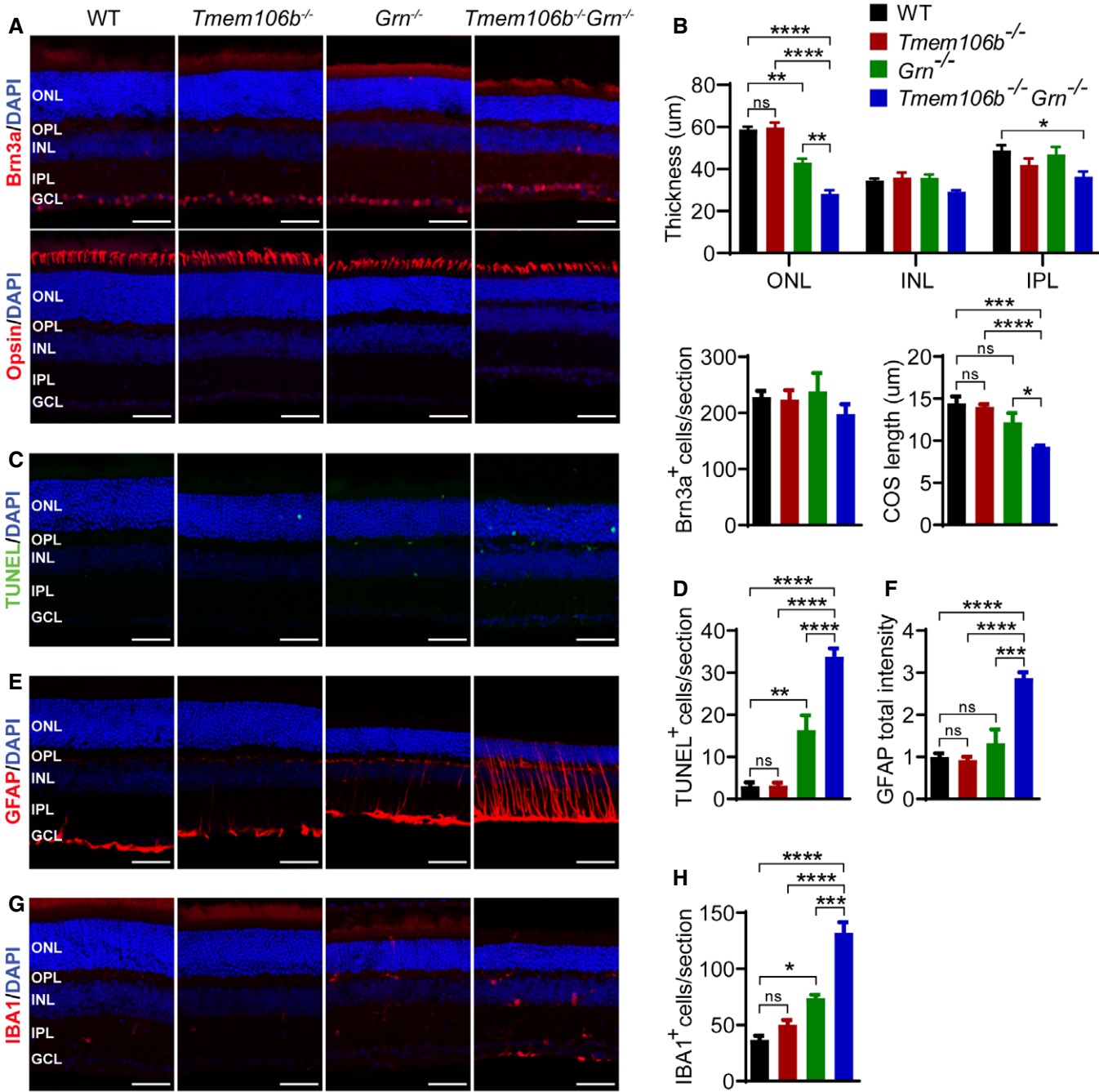

**Figure 5. Retinal neurodegeneration and gliosis in *Tmem106b⁻/⁻Grn⁻/⁻* mice.**

A, B   Retinal sections from 5-month-old mice were stained with indicated antibodies. Thickness of each layer, number of Brn3a-positive cells (retinal ganglion cells) per section and length of outer segment as indicated by opsin staining are quantified.

C, D   Increase in apoptosis in *Grn⁻/⁻* and *Tmem106b⁻/⁻Grn⁻/⁻* retina as indicated by TUNEL assay. TUNEL-positive signals per section were quantified.

E, F   Increase in Müller glial activation in *Tmem106b⁻/⁻Grn⁻/⁻* retina as indicated by GFAP staining. GFAP intensity per section was quantified and normalized to WT control.

G, H   Increase in microglia activation in *Grn⁻/⁻* and *Tmem106b⁻/⁻Grn⁻/⁻* retina as indicated by IBA1 staining. The number of IBA1-positive cells per section was quantified.

Data information: GCL, ganglion cell layer; INL, inner nuclear layer; IPL, inner plexiform layer; ONL, outer nuclear layer; OPL, outer plexiform layer. $n = 3$–6. Data presented as mean ± SEM. One-way ANOVA tests with Bonferroni's multiple comparisons: *$P < 0.05$, **$P < 0.01$, ***$P < 0.001$, ****$P < 0.0001$. Scale bar = 50 μm.

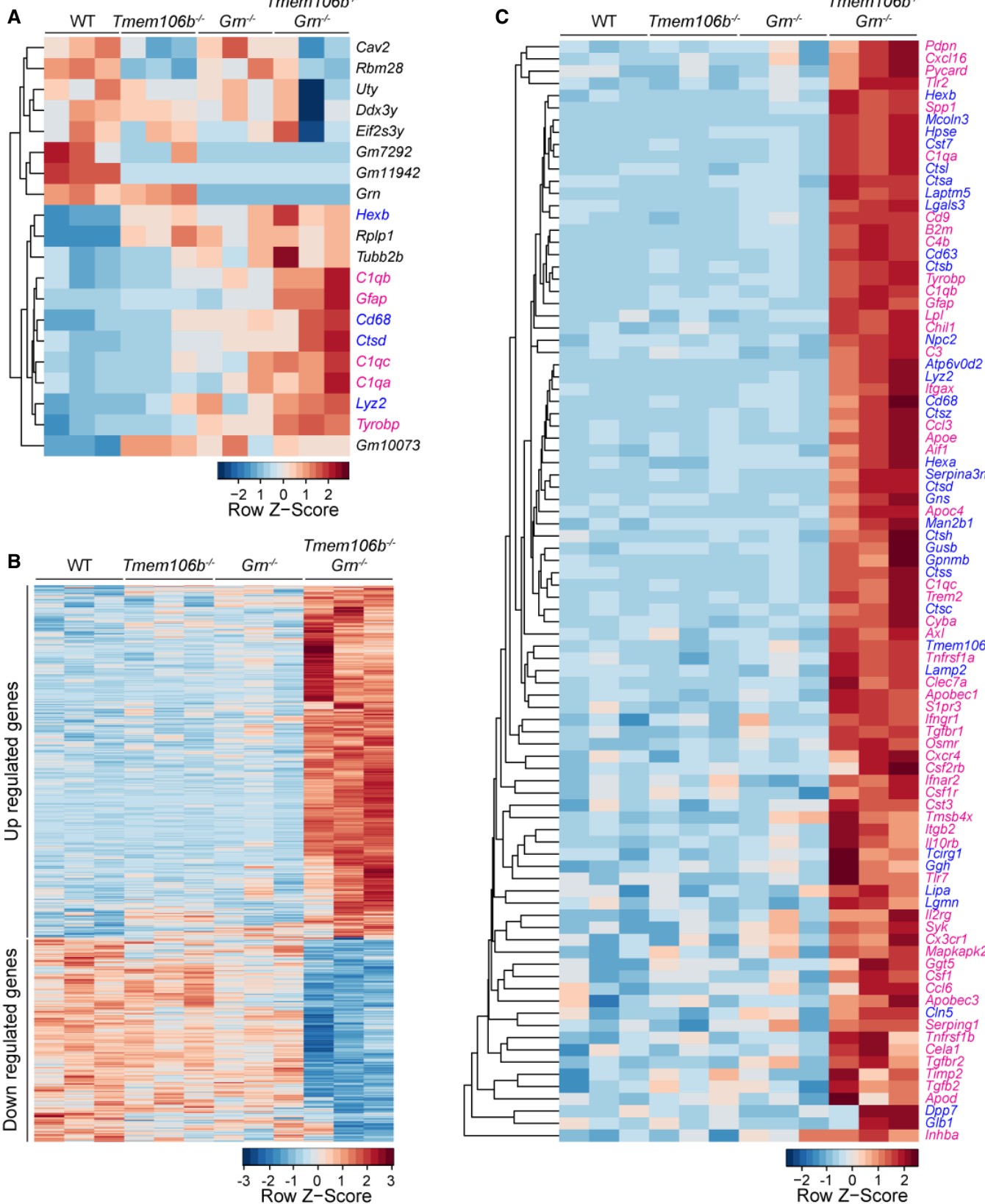

**Figure 6.**

**Figure 6.  Transcriptome analysis reveals increased expression of inflammation and lysosome genes in *Tmem106b$^{-/-}$Grn$^{-/-}$* mice.**

A, B    Heatmap illustrating gene expression changes in WT, *Tmem106b$^{-/-}$*, *Grn$^{-/-}$*, and *Tmem106b$^{-/-}$Grn$^{-/-}$* cortex (A) and spinal cord (B) samples isolated from 2.7-month-old mice. *n* = 3. Genes with FDR *P* ≤ 0.05, LogFC ≥ 0.5 or ≤ 0.5 between WT and *Tmem106b$^{-/-}$Grn$^{-/-}$* groups are plotted based on hierarchical clustering analysis.

C       Heatmap illustrating gene expression changes in the inflammation and lysosome pathways in WT, *Tmem106b$^{-/-}$*, *Grn$^{-/-}$*, and *Tmem106b$^{-/-}$Grn$^{-/-}$* spinal cord samples.

Data information: Inflammation genes are highlighted in magenta, and lysosomal genes are highlighted in blue.

Our RNA-Seq analysis revealed changes in the expression of several microglia and astrocyte-specific genes. To determine lysosome abnormalities in these cell types, we performed the co-staining of LAMP1, cathepsin D (CathD), and glial markers GFAP and IBA1. We found severe lysosome abnormalities in astrocytes and microglia in the spinal cord sections of *Tmem106b$^{-/-}$Grn$^{-/-}$* mice (Fig 10A and B). A significant increase in the staining intensities of lysosomal proteins, such as LAMP1 and cathepsin D, was detected in these cells. In addition, LAMP1-positive lysosomes are abnormally enlarged in both astrocytes and microglia in the *Tmem106b$^{-/-}$Grn$^{-/-}$* mice (Fig 10A and B), supporting that PGRN and TMEM106B are critical for lysosomal function in these cells.

To further examine the lysosomal dysfunction in *Tmem106b$^{-/-}$Grn$^{-/-}$* mice, ultrastructural analysis of spinal cord sections using transmission electron microscopy (TEM) was carried out. The TEM images showed the accumulation of electron-dense lysosomes and autophagosomes in myelinated axons (Fig 11A), indicating that the loss of PGRN and TMEM106B likely results in trafficking defects of autophagosomes and lysosomes in myelinated axons in addition to lysosome abnormalities in the AIS. From the TEM analysis, we also observed accumulation of myelin debris and dark amorphous inclusions in microglia (Fig 11B), indicating a defect in cellular clearance in the *Tmem106b$^{-/-}$Grn$^{-/-}$* microglia. The accumulation of autophagosomes in the double-knockout mice was further confirmed by Western blot analysis using antibodies against LC3, an autophagosome marker. A significant increase in the lipidated form of LC3 (LC3-II), which decorates autophagosome membranes, was observed in *Tmem106b$^{-/-}$Grn$^{-/-}$* spinal cord lysates (Fig 11C). This indicates an increase in autophagy induction or a block in autophagy–lysosome fusion/lysosomal degradation, with the latter more likely given the lysosomal abnormalities in *Tmem106b$^{-/-}$Grn$^{-/-}$* mice.

Many lysosomal genes are under the transcriptional control of members of the melanocyte inducing transcription factor (MiTF) family, including MiTF, TFEB, and TFE3. Under starvation or lysosomal stress, these transcriptional factors are translocated into the nucleus to induce the expression of many genes in the autophagy–lysosome pathway (Napolitano & Ballabio, 2016; Bajaj *et al*, 2019). To determine whether this pathway is activated in the *Tmem106b$^{-/-}$Grn$^{-/-}$* mice, we stained spinal cord sections with antibodies against TFE3. A significant increase of nuclear TFE3 signals was observed in

the microglia of the spinal cord sections of *Tmem106b$^{-/-}$Grn$^{-/-}$* mice, in comparison with those of *Tmem106b$^{-/-}$*, *Grn$^{-/-}$*, and WT controls (Fig 11D and E). This result indicates that lysosomal dysfunction in microglia results in a positive feedback loop through the TFE3/TFEB pathway to upregulate the expression of lysosomal genes. Activation of TFE3/TFEB could also drive the expression of inflammatory genes, which has been shown in several studies (Nabar & Kehrl, 2017; Brady *et al*, 2018).

### ALS/FTLD-related phenotypes in *Tmem106b$^{-/-}$Grn$^{-/-}$* mice

The accumulation of ubiquitin-positive aggregates comprised of TDP-43 is a hallmark of ALS/FTLD. However, these pathological features were not present in our PGRN-deficient mice (Fig 12A and B). No obvious TDP-43 pathology was observed in the 5-month-old TMEM106B-deficient mice either, although there is a small but significant increase in p62 levels in the insoluble fraction (Fig 12A and B). However, an obvious and highly significant accumulation of ubiquitinated proteins and autophagy adaptor protein p62 in both soluble and insoluble fractions and phosphorylated TDP-43 (S403/S404) in the insoluble fraction was detected in the 5-month-old *Tmem106b$^{-/-}$Grn$^{-/-}$* brain lysates and spinal cord lysates (Fig 12A and B). Immunostaining using antibodies against p62 and ubiquitin (Ub) showed accumulation of p62 and ubiquitin-positive aggregates in *Tmem106b$^{-/-}$Grn$^{-/-}$* spinal cord sections (Fig 12C and D). p62-positive aggregates are also present in the cortex, thalamus, hippocampus (Fig EV4), and cerebellum (Fig EV5) in the *Tmem106b$^{-/-}$Grn$^{-/-}$* mice. Taken together, these data support that ablation of TMEM106B enhances the manifestation of FTLD phenotypes in the PGRN-deficient background.

## Discussion

### TMEM106B is a critical regulator of lysosomal trafficking in motor neuron axons

TMEM106B has been shown to regulate many aspects of lysosomal function, including lysosome morphology (Chen-Plotkin *et al*, 2012; Lang *et al*, 2012; Brady *et al*, 2013), lysosome pH (Chen-

**Figure 7.  Increases in microglia activation in *Tmem106b$^{-/-}$Grn$^{-/-}$* mice.**

A–D    Western blot analysis of galectin-3 and GPNMB in the spinal cord (A, B) and brain (C, D) lysates from 5-month-old WT, *Tmem106b$^{-/-}$*, *Grn$^{-/-}$*, and *Tmem106b$^{-/-}$Grn$^{-/-}$* mice. *n* = 3. Data presented as mean ± SEM. One-way ANOVA tests with Bonferroni's multiple comparisons: ***P < 0.001, ****P < 0.0001.

E       Immunostaining of galectin-3, CathD, and CD68 in spinal cord sections from 5-month-old WT, *Tmem106b$^{-/-}$*, *Grn$^{-/-}$*, and *Tmem106b$^{-/-}$Grn$^{-/-}$* mice. Scale bar = 10 μm.

Source data are available online for this figure.

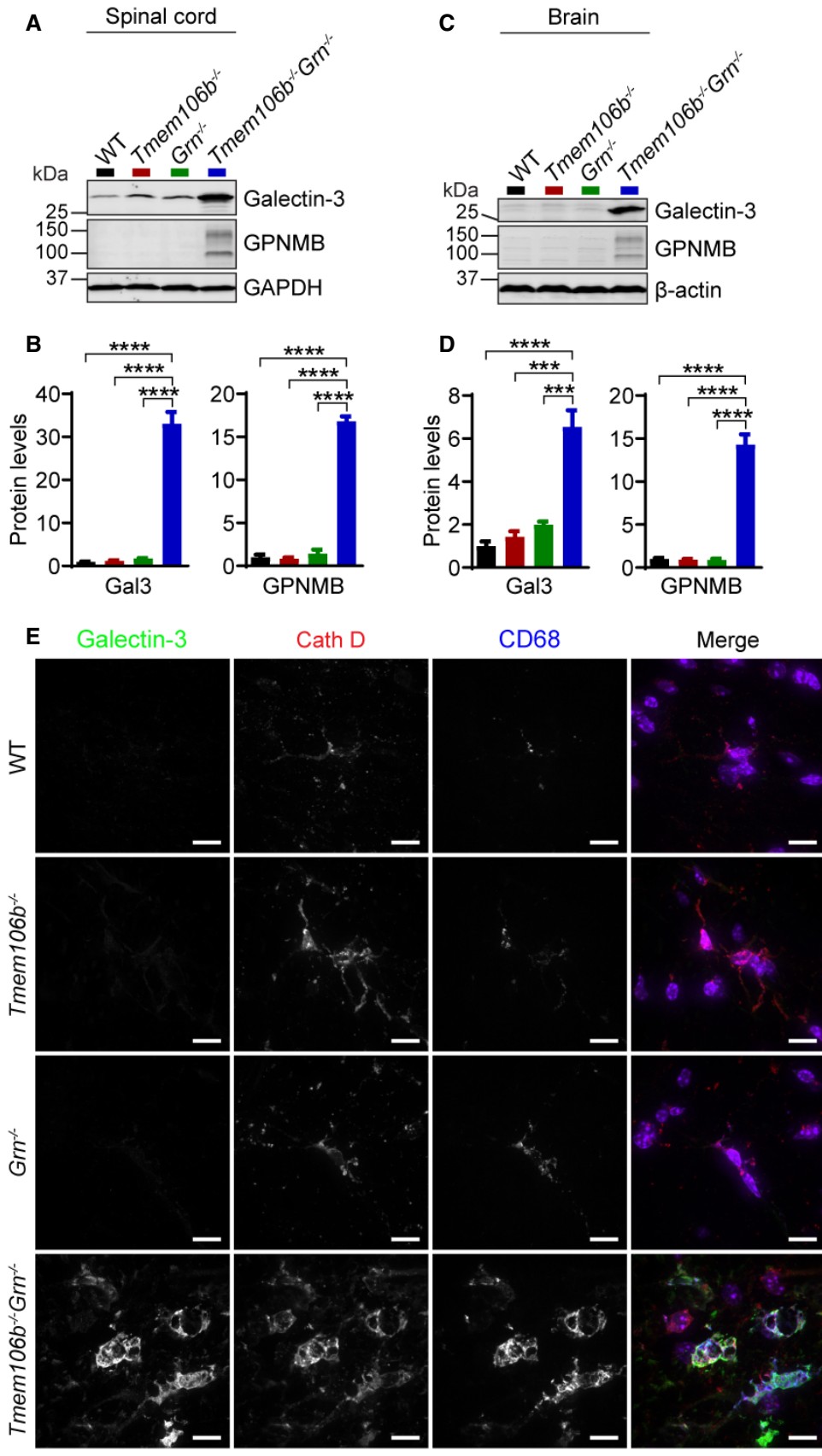

**Figure 7.**

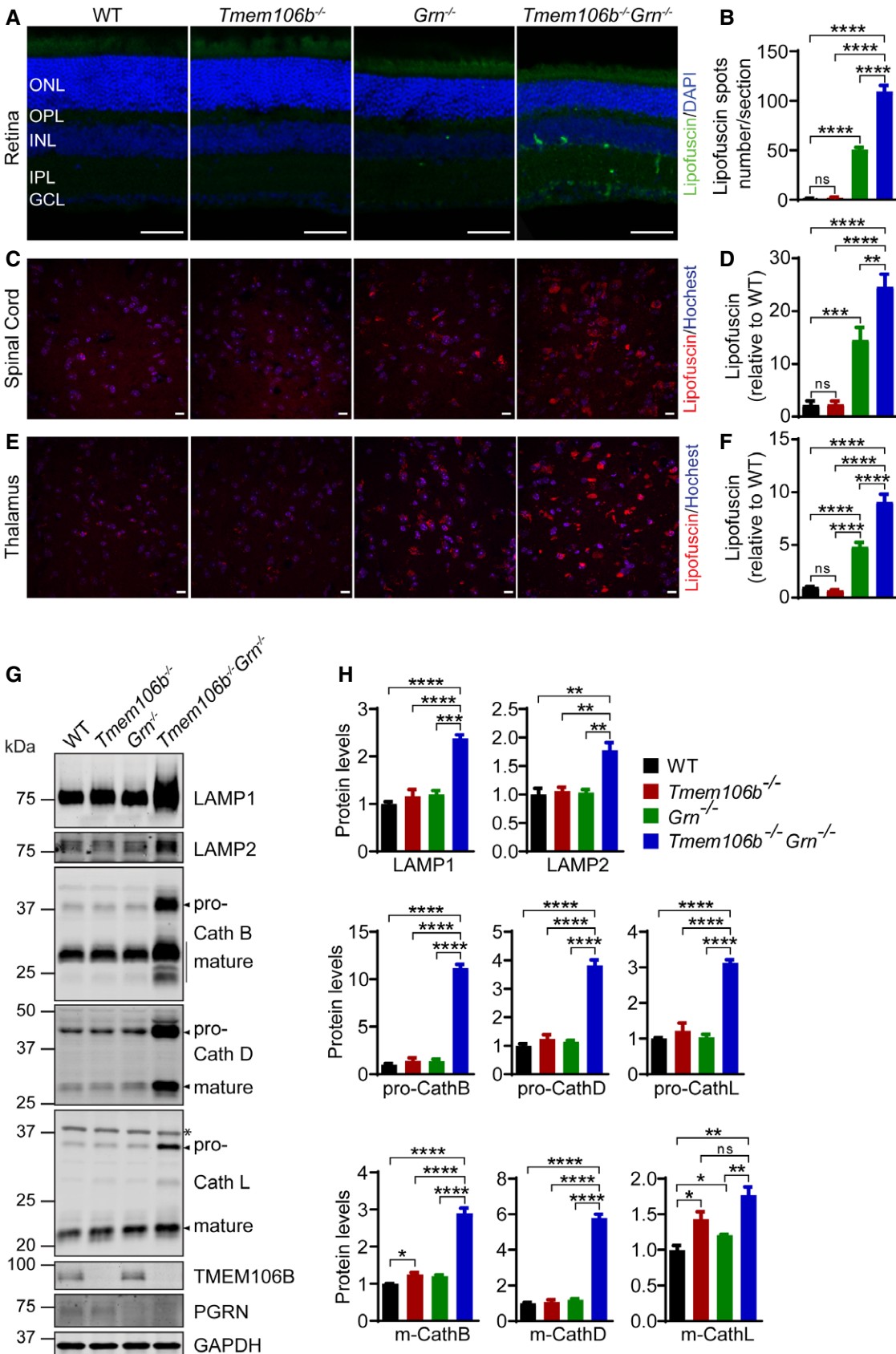

**Figure 8.**

**Figure 8.  Increases in the levels of lipofuscin and lysosome proteins in *Tmem106b⁻/⁻Grn⁻/⁻* mice.**

A, B     Lipofuscin accumulation in the retina: The number of autofluorescent puncta per section was quantified. ONL: outer nuclear layer; INL: inner nuclear layer; OPL: outer plexiform layer; IPL: inner plexiform layer; GCL: ganglion cell layer. *n* = 3–6. Scale bar = 50 μm.

C–F     TMEM106B and PGRN deletion results in increased autofluorescence in the spinal cord (C, D) and thalamus (E, F) of 5-month-old mice. *n* = 3. Scale bar = 10 μm.

G, H    Western blot analysis of lysosomal proteins in the spinal cord lysates from 5-month-old WT, *Tmem106b⁻/⁻*, *Grn⁻/⁻*, and *Tmem106b⁻/⁻Grn⁻/⁻* mice. Asterisk indicates non-specific bands. *n* = 3.

Data information: For all the analyses, data are presented as mean ± SEM. One-way ANOVA tests with Bonferroni's multiple comparisons: *$P < 0.05$; **$P < 0.01$; ***$P < 0.001$; ****$P < 0.0001$; ns, no significance.

Source data are available online for this figure.

Plotkin *et al*, 2012; Klein *et al*, 2017; Kundu *et al*, 2018), lysosome exocytosis (Kundu *et al*, 2018), and lysosomal trafficking in neuronal dendrites (Schwenk *et al*, 2014). In this study, we have found that TMEM106B deficiency leads to the accumulation of LAMP1 and cathepsin D-positive vacuoles at the distal end of the AIS of motor neurons in the spinal cord (Fig 1A and B), consistent with a recent report (Luningschror *et al*, 2020). This result supports a critical role of TMEM106B in regulating lysosomal trafficking and/or dynamics at the distal end of the AIS. Although the detailed mechanism remains to be determined, TMEM106B was shown to regulate retrograde lysosomal transport in motor neurons (Luningschror *et al*, 2020). The selective effect of TMEM106B loss on motor neurons is puzzling, given that TMEM106B has been identified as a FTLD risk factor. It is possible that long axons and the complex arborization of motor neurons render them more susceptible to any subtle perturbation. In our recent study, we found that TMEM106B deficiency leads to myelination defects (Feng *et al*, 2020). Although myelination defects not necessarily affect the integrity of the AIS (Clark *et al*, 2016), it is possible that a slight alteration of myelin sheath formation in TMEM106B-deficient motor neurons makes the axon membrane easy to swell at the distal end of the AIS, where myelination begins. We speculate that the myelination defects together with increased retrograde lysosomal trafficking and possibly lysosome fusion events result in the formation of giant lysosome vacuoles specifically at the distal end of the AIS.

Since lysosomes are required for the degradation of autophagic cargoes, lysosomal dysfunction often leads to a defect in autophagy flow. We have detected the accumulation of p62 and ubiquitinated proteins in the brain and spinal cord lysates of 16-month-old TMEM106B-deficient mice (Fig 1C–F), indicating that lysosomal dysfunction caused by the loss of TMEM106B leads to autophagy defects during aging. Similar phenotypes have been reported in a recently published study (Luningschror *et al*, 2020), although at a much younger age.

**Ablation of TMEM106B exacerbates phenotypes associated with loss of PGRN**

In direct contradiction to a recently published report (Klein *et al*, 2017), we found that mice deficient in both TMEM106B and PGRN develop severe neurodegenerative phenotypes, with reduced motor activity in the open-field test (Fig 2B and C), hindlimb weakness, and altered clasping behavior (Fig 2D and E). These mice develop severe ataxia around 5 months of age (Movies EV1 and EV2) and must be sacrificed. Severe neuronal loss and microglia and astrocyte activation were observed in the spinal cord (Fig 3), brain (Fig 4), and retina (Fig 5) at 5 months of age. Our TMEM106B-deficient

mice generated using the CRISPR/Cas9 technique completely removed any TMEM106B products and have been backcrossed with wild-type C57/BL6 mice for seven generations to eliminate any CRISPR off-target effects before crossing with *Grn⁻/⁻* mice, thus allowing us to re-test the genetic interaction between PGRN and TMEM106B in mouse models. It should be noted that two additional studies published in this issue of EMBO reports have also found similar phenotypes in independently created *Tmem106b⁻/⁻Grn⁻/⁻* mice (Werner *et al*, 2020; Zhou *et al*, 2020a). Together, our results strongly argue that a TMEM106B deficiency exacerbates phenotypes associated with loss of PGRN.

**Lysosomal abnormalities and autophagy defects in *Tmem106b⁻/⁻Grn⁻/⁻* mice**

Analysis of lysosomal phenotypes in the *Tmem106b⁻/⁻Grn⁻/⁻* mice showed severe lysosome abnormalities, indicated by a significant increase in lipofuscin signals (Fig 8A–F); the upregulation of many lysosomal proteins shown by Western blot analysis (Figs 8G and H, and EV3) and immunostaining (Figs 7E and 10), lysosome enlargement in astrocytes (Fig 10A) and microglia (Fig 10B) and the accumulation of lysosomes shown in TEM images (Fig 11A). These phenotypes are reminiscent of features exhibited by lysosome storage diseases and further corroborate the critical roles of PGRN and TMEM106B in the lysosome. In addition, *Tmem106b⁻/⁻ Grn⁻/⁻* mice show a defect in autophagy with a significant increase in LC3-II levels and accumulation of autophagosomes (Fig 11), which likely explains the accumulation of p62 and ubiquitinated proteins in the brain lysates and spinal cord lysates of these mice (Fig 12).

While the exact function of PGRN in the lysosome is still unclear, it has been shown that PGRN is processed to granulin peptides in the lysosomal compartment (Holler *et al*, 2017; Lee *et al*, 2017; Zhou *et al*, 2017b). The loss of PGRN leads to reduced cathepsin D activities (Beel *et al*, 2017; Valdez *et al*, 2017; Zhou *et al*, 2017a; Butler *et al*, 2019), defects in lysosomal trafficking of prosaposin (Zhou *et al*, 2017c), and reduced glucocerebrosidase activities (Arrant *et al*, 2019; Valdez *et al*, 2019; Zhou *et al*, 2019). In mouse models, PGRN deficiency has been shown to result in lysosomal dysfunction in an age-dependent manner (Kao *et al*, 2017; Paushter *et al*, 2018). In aged *Grn⁻/⁻* mice, lipofuscin deposits and enlarged lysosomes were observed, as well as diffuse or granular cytosolic ubiquitin deposits and p62 aggregation (Ahmed *et al*, 2010; Yin *et al*, 2010b; Wils *et al*, 2012; Tanaka *et al*, 2014). Reduced autophagic flux and autophagy-dependent clearance have also been reported in PGRN-deficient mice (Chang *et al*, 2017). Early lysosomal deficits have been reported specifically in PGRN-deficient

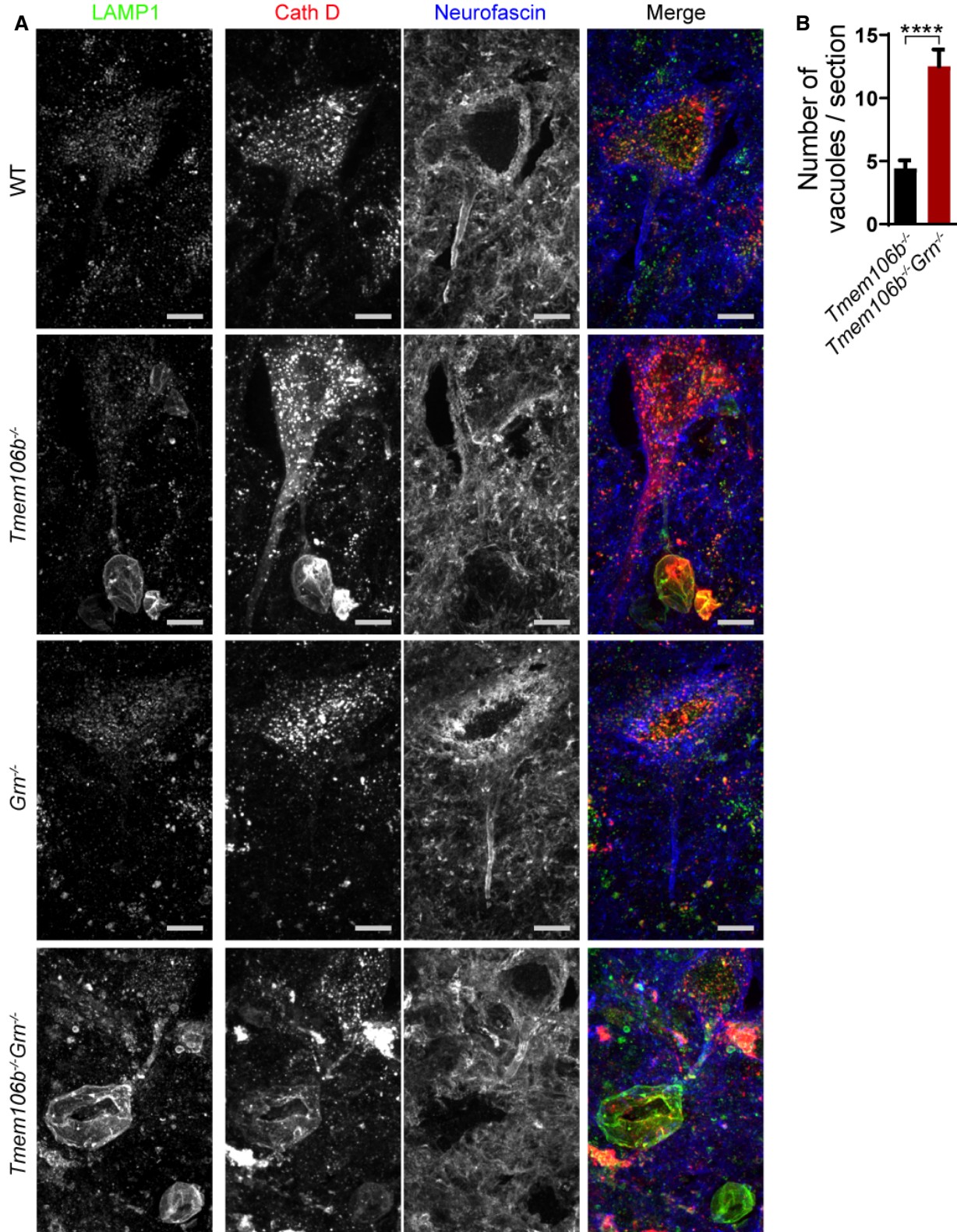

**Figure 9. Lysosomal vacuolization in the axon initial segments of motoneurons in spinal cord sections of *Tmem106b*$^{-/-}$ and *Tmem106b*$^{-/-}$*Grn*$^{-/-}$ mice.**

A  Spinal cord sections from 5-month-old WT, *Tmem106b*$^{-/-}$, *Grn*$^{-/-}$, and *Tmem106b*$^{-/-}$*Grn*$^{-/-}$ mice were stained with anti-neurofascin, anti-LAMP1, and anti-CathD antibodies. Representative images from three independent mice were shown. Scale bar = 10 µm.

B  Quantification of LAMP1-positive vacuoles (> 2 µm) for experiments in (A). Four–six sections/spinal cord from three mice were analyzed for each genotype. Data presented as mean ± SEM. Unpaired Student's *t*-test: ****$P$ < 0.0001.

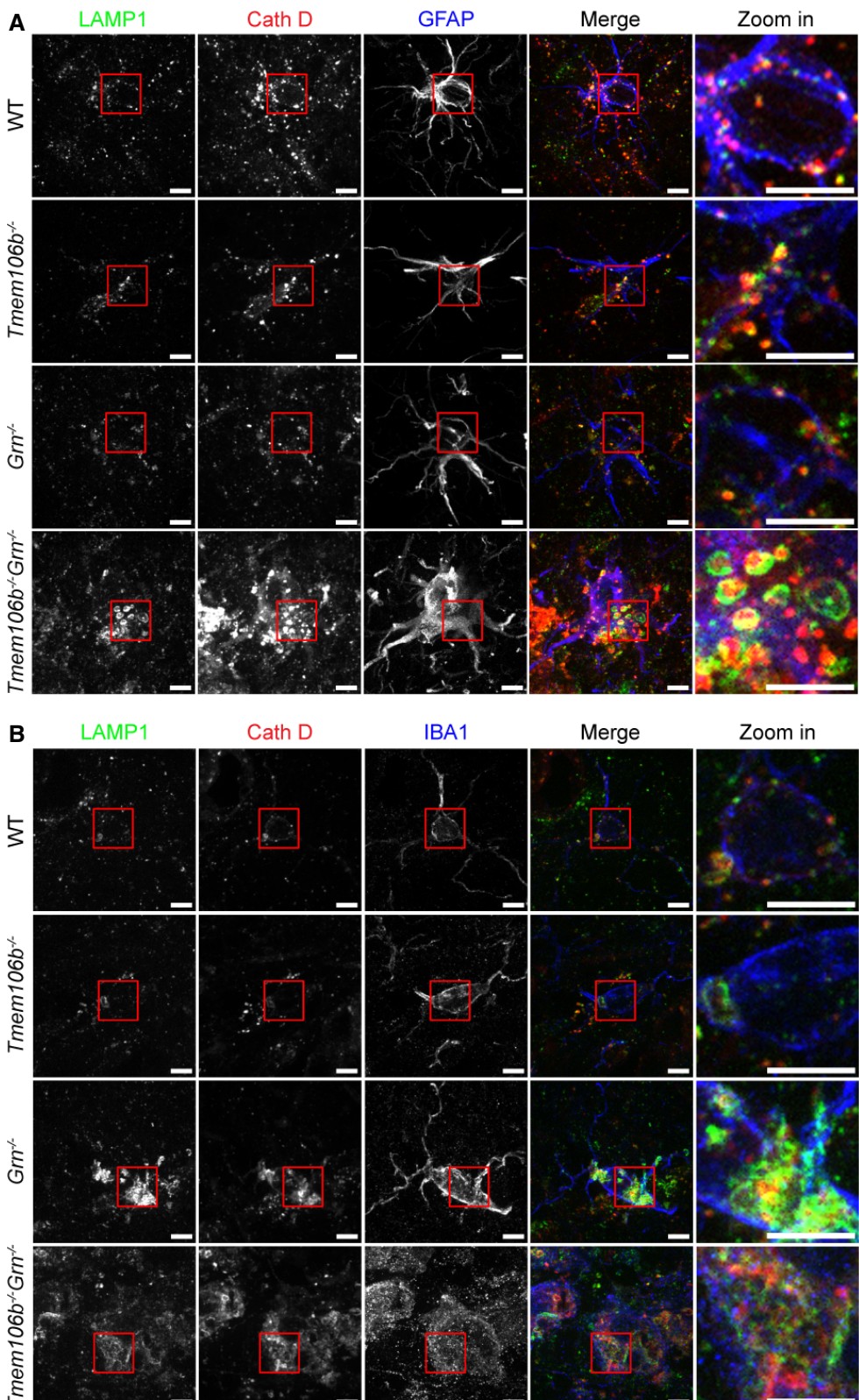

**Figure 10. Lysosome abnormalities in *Tmem106b$^{-/-}$Grn$^{-/-}$* astrocytes and microglia.**

A, B   Immunostaining of GFAP (A) or IBA1 (B) with LAMP1 and Cath D in the spinal cord sections from 5-month-old WT, *Tmem106b$^{-/-}$*, *Grn$^{-/-}$*, and *Tmem106b$^{-/-}$Grn$^{-/-}$* mice. Scale bar = 5 μm.

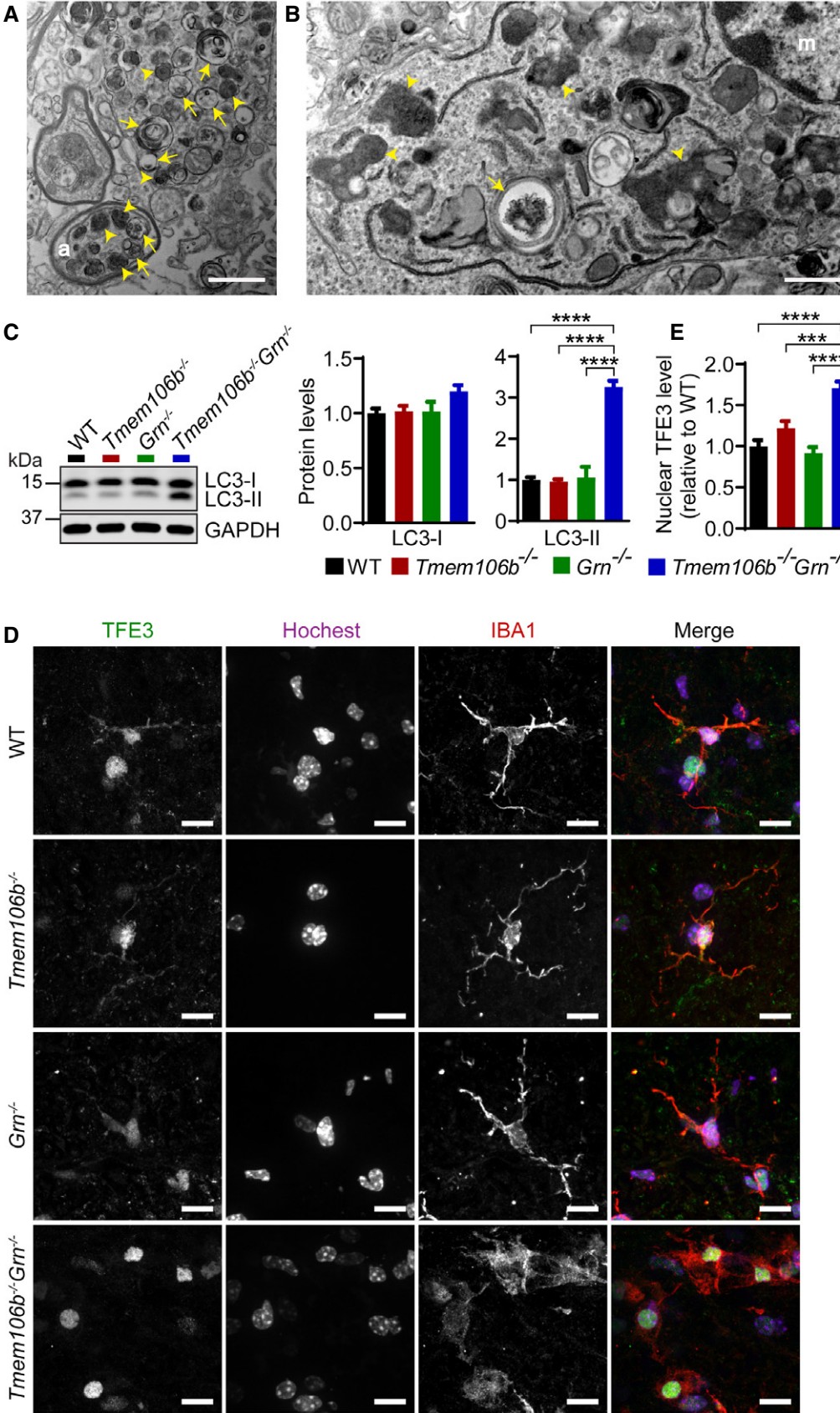

**Figure 11.**

**Figure 11.   Autophagy–lysosome dysfunction in the *Tmem106b$^{-/-}$Grn$^{-/-}$* spinal cord.**

A    Accumulation of autophagosomes and lysosomes in the spinal cord sections from a 5-month-old *Tmem106b$^{-/-}$Grn$^{-/-}$* mouse as shown by transmission electron microscope analysis. Scale bar = 1 μm. Arrows: autophagosome; arrowheads: lysosomes; (a): axon.

B    Accumulation of myelin debris and amorphous inclusions in the microglia in the spinal cord sections from a 5-month-old *Tmem106b$^{-/-}$Grn$^{-/-}$* mouse as shown by transmission electron microscope analysis. Scale bar = 1 μm. Arrows: myelin debris; arrowheads: dark amorphous inclusions; (m): microglia.

C    Western blot analysis of LC3-I and LC3-II in the spinal cord lysates from 5-month-old WT, *Tmem106b$^{-/-}$*, *Grn$^{-/-}$*, and *Tmem106b$^{-/-}$Grn$^{-/-}$* mice. *n* = 3. Data presented as mean ± SEM. One-way ANOVA tests with Bonferroni's multiple comparisons: ****$P < 0.0001$.

D, E    Immunostaining of TFE3 and IBA1 in the spinal cord sections from 5-month-old WT, *Tmem106b$^{-/-}$*, *Grn$^{-/-}$*, and *Tmem106b$^{-/-}$Grn$^{-/-}$* mice. Nuclear signals of TFE3 in IBA1-positive microglia were quantified. *n* = 3. Data presented as mean ± SEM. One-way ANOVA tests with Bonferroni's multiple comparisons: ***$P < 0.001$, ****$P < 0.0001$. Scale bar = 10 μm.

Source data are available online for this figure.

microglia (Gotzl *et al*, 2018). PGRN deficiency also leads to increased lysosome exophagy in macrophages, which enhances atherosclerosis (Nguyen *et al*, 2018).

Although the loss of either PGRN or TMEM106B alone results in lysosomal dysfunction in mouse models, the lysosome abnormalities are exacerbated in *Tmem106b$^{-/-}$Grn$^{-/-}$* mice and appear at a much earlier stage. At 2.7 months of age, there are minimal changes in gene expression in *Tmem106b$^{-/-}$* or *Grn$^{-/-}$* mice. However, massive upregulation of lysosomal and inflammatory genes was observed in 2.7-month-old *Tmem106b$^{-/-}$Grn$^{-/-}$* spinal cord samples as shown in our RNA-Seq analysis (Figs 6 and EV2). With the non-overlapping functions of PGRN and TMEM106B in the autophagy–lysosome pathway, it is not surprising that deficiency of both proteins leads to severe defects.

### Neurodegenerative phenotypes in *Tmem106b$^{-/-}$Grn$^{-/-}$* mice

The severe hindlimb weakness and motor dysfunction observed in the *Tmem106b$^{-/-}$Grn$^{-/-}$* mice led us to examine pathologies in the spinal cord. A significant reduction of NeuN-positive neurons and CHAT-positive motor neurons was found in the *Tmem106b$^{-/-}$Grn$^{-/-}$* spinal cord sections (Fig 3). Motor neurons are likely more affected likely due to a specific role of TMEM106B in regulating lysosome dynamics in the AIS region of motor neurons (Fig 1A and B) (Luningschror *et al*, 2020). In the brain, there is also a significant reduction of NeuN protein levels in *Tmem106b$^{-/-}$Grn$^{-/-}$* mice, although to a lesser extent compared to that of the spinal cord (Fig 4). It is possible that specific subsets of neurons in the brain are more sensitive to the loss of PGRN and TMEM106B and future studies using specific markers might help us identify the subtypes of neurons that are affected. Since PGRN and TMEM106B are associated with FTLD in humans, identification of the types of neurons affected by these two genes in the regions affected by FTLD might give insights into the selective vulnerability of neurons.

Accumulation of lysosomes and autophagosomes is observed in the myelinated axons in the spinal cord sections of *Tmem106b$^{-/-}$Grn$^{-/-}$* mice (Fig 11A), indicating that the transport of lysosomes and autophagosomes might be impaired in *Tmem106b$^{-/-}$Grn$^{-/-}$* neurons. While the cause of this phenotype is unclear at the moment, defects in autophagosome and lysosomal trafficking in myelinated axons combined with lysosomal defects near the AIS of motor neurons and reduced lysosomal enzyme activities due to the loss of PGRN are likely to drive neuronal dysfunction and eventually result in neuronal death in the double-knockout mice.

### Glial dysfunction in *Tmem106b$^{-/-}$Grn$^{-/-}$* mice

Progranulin deficiency has been shown to cause microglial activation in an age-dependent manner (Martens *et al*, 2012; Lui *et al*, 2016; Kao *et al*, 2017; Takahashi *et al*, 2017; Gotzl *et al*, 2019). At 2.7 months of age, there are minimal gene expression changes in PGRN-deficient brain or spinal cord samples in our RNA-Seq analysis (Fig 6). However, massive upregulation of inflammatory genes has been observed in the spinal cord and to a lesser extent, in the brain samples of the *Tmem106b$^{-/-}$Grn$^{-/-}$* mice, indicating that the loss of TMEM106B exacerbates microglial activation in PGRN-deficient mice. Examination of lysosome abnormalities revealed enlarged lysosomes and elevated levels of lysosomal proteins in microglia and astrocytes (Fig 10). The lysosomal phenotypes in astrocyte and microglia could be attributed to both lysosomal dysfunction in these cells and an increased load to clear neuronal debris due to neuronal death in the *Tmem106b$^{-/-}$Grn$^{-/-}$* mice. It is likely that lysosomal dysfunction in microglia and astrocytes further triggers mis-regulated activation of these cells, which aggravates neurodegeneration in the *Tmem106b$^{-/-}$Grn$^{-/-}$* mice.

In addition, a single point mutation in the lumenal domain of TMEM106B has recently been associated with hypo-myelinating leukodystrophy (Simons *et al*, 2017; Yan *et al*, 2018). Myelination defects have been observed in TMEM106B-deficient mice (Feng *et al*, 2020; Zhou *et al*, 2020b). While the extent of myelin disruption remains to be tested in the *Tmem106b$^{-/-}$Grn$^{-/-}$* mice, myelinating defects might also contribute to the pathology of the *Tmem106b$^{-/-}$Grn$^{-/-}$* mice.

### The role of TMEM106B in FTLD

Although PGRN haploinsufficiency is a leading cause of FTLD with ubiquitin-positive TDP-43 aggregates in humans (Baker *et al*, 2006; Cruts *et al*, 2006; Gass *et al*, 2006; Neumann *et al*, 2006), PGRN deficiency in mice does not result in robust accumulation of ubiquitinated proteins or TDP-43 aggregates in our studies (Fig 12). Since TMEM106B has been identified as the main risk factor for FTLD-*GRN*, genetic manipulation of TMEM106B levels might allow us to generate a better mouse model for FTLD-*GRN* based on the genetic interaction between these two genes in FTLD.

Since increased TMEM106B protein levels correlate with increased risks for FTLD with *GRN* mutations, we have generated TMEM106B transgenic mice and found that TMEM106B overexpression in neurons exacerbates lysosomal abnormalities caused

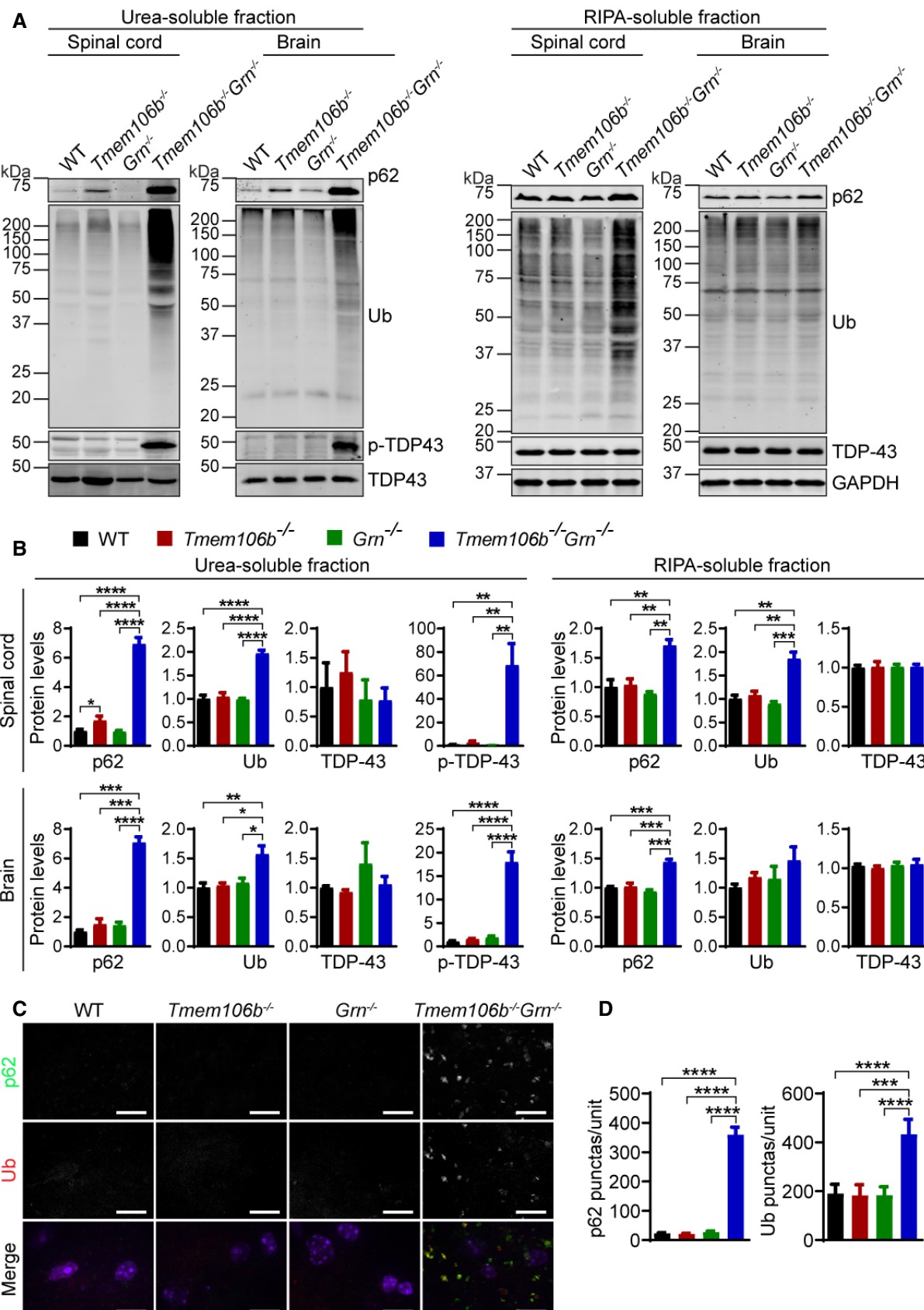

Figure 12.

**Figure 12.  FTLD-related pathological changes in *Tmem106b$^{-/-}$ Grn$^{-/-}$* mice.**

A, B   Western blot analysis of p62, ubiquitin (Ub), TDP-43 and p-TDP-43 in RIPA- and urea-soluble fractions from spinal cord (C5–C8) and brain of 5-month-old WT, *Tmem106b$^{-/-}$*, *Grn$^{-/-}$*, and *Tmem106b$^{-/-}$ Grn$^{-/-}$* mice. Spinal cord: *n* = 3; brain: *n* = 4. Data presented as mean ± SEM. One-way ANOVA tests with Bonferroni's multiple comparisons: *$P < 0.05$; **$P < 0.01$, ***$P < 0.001$, ****$P < 0.0001$.

C, D   Immunostaining of p62 and ubiquitin in the spinal cord sections from 5-month-old WT, *Tmem106b$^{-/-}$*, *Grn$^{-/-}$*, and *Tmem106b$^{-/-}$ Grn$^{-/-}$* mice. The number of p62 or Ub positive puncta was quantified. *n* = 3. Data presented as mean ± SEM. One-way ANOVA tests with Bonferroni's multiple comparisons: ***$P < 0.001$, ****$P < 0.0001$. Scale bar = 10 μm.

Source data are available online for this figure.

by the loss of PGRN (Zhou *et al*, 2017d), consistent with their interaction as observed in human FTLD cases. In this study, we further showed that ablation of TMEM106B in a PGRN deficienct background leads to severe lysosomal abnormalities and neurodegeneration. Several pathological features of FTLD, including accumulation of autophagy adaptor protein p62, ubiquitinated protein and LC3 were observed in our *Tmem106b$^{-/-}$* mice at 16 months of age (Fig 1C–F). Phosphorylated TDP-43, a hallmark of FTLD, was also detected in the spinal cord and brain lysates of 16-month-old TMEM106B-deficient mice (Fig 1C–F). These FTLD-related pathologies were evident in the brain and spinal cord of 5-month-old *Tmem106b$^{-/-}$ Grn$^{-/-}$* mice but not *Grn$^{-/-}$* mice (Fig 12). These data strongly indicate that the loss of function of TMEM106B potentiates the development of FTLD pathology. Taken together, our studies suggest that TMEM106B levels need to be tightly regulated. Either too much or too little TMEM106B will lead to the disruption of proper lysosomal function as shown both *in vitro* in cell culture and *in vivo* in mouse models, which can have detrimental effects on the health of neurons.

In that regard, it will be interesting to analyze the phenotypes of *Tmem106b$^{+/-}$ Grn$^{-/-}$* mice in order to determine whether a partial loss of function of TMEM106B alters phenotypes associated with PGRN deficiency. Since FTLD is caused by PGRN haploinsufficiency in humans, it is also important to examine phenotypes in *Tmem106b$^{-/-}$ Grn$^{+/-}$* and *Tmem106b$^{+/-}$ Grn$^{+/-}$* mice in order to determine whether we can utilize the genetic interaction between TMEM106B and PGRN to establish a better mouse model for FTLD-*GRN*. Although no obvious motor deficits or ataxia phenotypes have been observed in these mice, more work in the future is required to fully characterize molecular and cellular pathological changes in these mice.

Our studies further underscore the critical roles of PGRN and TMEM106B in the lysosome in maintaining the health of the nervous system. Besides FTLD, *GRN* and *TMEM106B* have been implicated in brain aging (Rhinn & Abeliovich, 2017) and many other neurodegenerative diseases. In addition to *GRN* and *TMEM106B*, many other lysosomal genes are directly involved in neurodegenerative diseases and lysosomal dysfunction has emerged as a common mechanism for neurodegeneration (Peng *et al*, 2019). The dosage effect seen in the *GRN* gene has been replicated in many other lysosomal genes, in which homozygous mutations lead to early-onset lysosomal storage disorders while heterozygous mutations lead to adult-onset neurodegenerative diseases. Future studies are needed to dissect the exact functions of PGRN, TMEM106B, and lysosomes in maintaining a healthy nervous system.

# Material and Methods

### Primary antibodies and reagents

The following antibodies were used in this study: mouse anti-GAPDH (Proteintech Group, 60004-1-Ig), rat anti-mouse LAMP1 (BD Biosciences, 553792), LAMP2 (Developmental Studies Hybridoma Bank, GL2A7-c), goat anti-CathB (R&D Systems, AF965), goat anti-CathD (R&D Systems, AF1029), goat anti-CathL (R&D Systems, AF1515), mouse anti-CHAT (R&D Systems, AF3447), mouse anti-NeuN (Millipore, MAB377), mouse anti-GFAP (GA5) (Cell signaling, 3670S), rabbit anti IBA-1 (Wako, 01919741), goat anti-IBA1 (Novus Biologicals, NB100-1028), rabbit anti-red/green opsin (Millipore, AB5405); goat anti-blue opsin (Santa Cruz, sc-14363), rabbit anti-Brn3a (Millipore, MAB1585), rabbit anti-Synaptophysin (Proteintech Group, 17785-1-AP), mouse anti-PSD95 (Millipore, MABN68), mouse anti-galectin-3 (BioLegend, 126702), goat anti-GPNMB (R&D Systems, AF2330), rat anti-CD68 (Bio-Rad, MCA1957), chicken anti-neurofascin (R&D Systems, AF3235), rabbit anti-TFE3 (Sigma, HPA023881), mouse anti-LC3 (Cell signaling, 83506S), rabbit anti-p62 (MBL, PM045), mouse anti-Ubiquitin (BioLegend, 646302), mouse anti-TDP43 (R&D Systems, MAB7778), rabbit anti-p-TDP-43 S403/S404 (COSMO BIO, CAC-TIP-PTD-P05), and sheep anti-mouse PGRN (R&D Systems, AF2557). Rabbit anti-TMEM106B antibodies were characterized previously (Brady *et al*, 2013).

The following reagents were also used in the study: TUNEL Bright-Green Apoptosis Detection Kit (Vazyme Biotech, A112), VECTAS-TAIN Elite ABC Kit (Vector Laboratories, PK-6200), DAB peroxidase substrate kit (Vector Laboratories, SK-4100), VECTAMOUNT permanent mounting medium (Vector laboratories, H-5000), Fluoromount-G (Thermo scientific, E113391), TrueBlack Lipofuscin Autofluorescence Quencher (Biotium, 23007), Odyssey blocking buffer (LI-COR Biosciences, 927-40000), protease inhibitor (Roche, 05056489001), Pierce BCA Protein Assay Kit (Thermo scientific, 23225), O.C.T compound (Electron Microscopy Sciences, 62550-01), and Quetol 651/NSA Embedding Kit (Electron Microscopy Sciences, 14640).

### Mouse strains

TMEM106B knockout mice were produced using CRISPR/Cas9 genome editing with two guide RNAs (5′-AGTGAAGTGCACAAC GAAGA-3′, 5′-ACCCTATGGGATATATTTAC-3′) targeting exon 3 of mouse TMEM106B, flanking the start codon. C57BL/6J × FVB/N mouse embryos were injected with gRNAs and Cas9 mRNA at the Cornell Transgenic Core Facility. Editing was confirmed by sequencing PCR products from genomic DNA, and the loss of protein products was determined by Western blot of tissue lysates. Offspring from

the founder containing 341 bp deletion were backcrossed to C57BL/6J for seven generations and used for the study. Δ341 bp knockout mouse genotyping was performed by PCR using oligos 5′-GTGCA-CAACGAAGACGGAAG-3′ and 5′-TGGCAAACCTCAGGCTCATT-3′ to specifically detect the WT allele (550 bp), and using oligos 5′-GTTCACCTGCAGTGCCAACT-3′ and 5′-TTGTTTGCTTTTGTGTTTC TGA-3′ to detect the mutant allele (397 bp). C57/BL6 and $Grn^{-/-}$ mice (Yin *et al*, 2010a) were obtained from The Jackson Laboratory. All animals (1–6 adult mice per cage) were housed in a 12 h light/dark cycle. Mixed male and female mice were used for this study.

## Behavioral test

4.5- to 5-month-old mixed male and female $Grn^{-/-}$, $Tmem106b^{-/-}$, and $Tmem106b^{-/-}Grn^{-/-}$ mice, and WT controls in the C57/BL6 background (8–10 mice/group) were subject to the following behavioral tests: (i) Open-field test: Mice were placed in a clear plastic chamber (40 × 40 × 40 cm) for 15 min. Total movements in the open field were automatically recorded by the Viewer III software (Biobserve, Bonn, Germany). The apparatus was thoroughly cleaned with 70% ethanol. (ii) Hindlimb clasping test: Briefly, mice were suspended by the base of the tail and their behaviors were recorded for 30 s by the Viewer III software (Biobserve, Bonn, Germany). Hindlimb clasping was rated from 1 to 3 based on severity (Lieu *et al*, 2013). For all behavioral analyses, experimenters were blind to the genotypes of the mice.

## Tissue preparation for Western blot analysis

Mice were perfused with 1× PBS, and tissues were dissected and snap-frozen with liquid nitrogen and kept at −80°C. On the day of the experiment, frozen tissues were thawed and homogenized on ice with bead homogenizer (Moni International) in ice-cold RIPA buffer (150 mM NaCl, 50 mM Tris–HCl [pH 8.0], 1% Triton X-100, 0.5% sodium deoxycholate, 0.1% SDS) with 1 mM PMSF, proteinase, and phosphatase inhibitors. After centrifugation at 14,000 $g$ for 15 min at 4°C, supernatants were collected as the RIPA-soluble fraction. The insoluble pellets were washed with RIPA buffer and extracted in 2× v/w of Urea buffer (7 M Urea, 2 M Thiourea, 4% CHAPS, 30 mM Tris, pH 8.5). After sonication, samples were centrifuged at 200,000 $g$ at 24°C for 1 h and the supernatant collected as the Urea-soluble fraction. Protein concentrations were determined via BCA assay and then standardized. Equal amounts of protein were analyzed by Western blotting using the indicated antibodies.

## Immunofluorescence staining, image acquisition, and analysis

Mice were perfused with cold PBS, and tissues were post-fixed with 4% paraformaldehyde. After dehydration in 30% sucrose buffer, tissues were embedded in O.C.T compound (Electron Microscopy Sciences). 15-μm-thick sections were blocked and permeabilized with either 0.1% saponin in Odyssey blocking buffer or 0.2% Triton X-100 in 1× PBS with 10% horse serum before incubating with primary antibodies overnight at 4°C. The next day, sections were incubated with secondary fluorescent antibodies at room temperature for 1 h. After fluorescence immunolabeling, the sections were stained with Hoechst and then mounted using mounting medium (Vector laboratories). To block the autofluorescence, all sections

were incubated with 1× TrueBlack Lipofuscin Autofluorescence Quencher (Biotium) in 70% ethanol for 30 s at room temperature before or after the staining process. Antigen retrieval was performed by microwaving the sections in sodium citrate buffer (pH 6.0) for 15 min. Images were acquired on a CSU-X spinning disk confocal microscope (Intelligent Imaging Innovations) with an HQ2 CCD camera (Photometrics) using 40×, 63×, and 100× objectives. Eight to 10 different random images were captured, and the fluorescence intensity was measured directly with ImageJ after a threshold application. Lower magnification images were captured by 10× or 20× objectives on a Leica DMi8 inverted microscope. Three to five images were captured from each sample, and the fluorescence intensity was measured directly with ImageJ after a threshold application. Data from ≥ 3 brains in each genotype were used for quantitative analysis.

## Immunohistochemistry

Spinal cord sections were pretreated with 10 mM sodium citrate buffer (pH 6.0) for 15 min for antigen retrieval. After washing with PBS, sections were incubated with 3% hydrogen peroxidase in PBS for 10 min at room temperature to quench the endogenous peroxidase activity. Sections were then blocked and permeabilized with 0.2% Triton X-100 in PBS with 10% horse serum followed by incubating with goat anti-CHAT antibody overnight at 4°C. After washing in 1× PBS, sections were incubated sequentially with biotinylated rabbit anti-goat secondary antibody, prepared VECTAS-TAIN Elite ABC reagent (Vector Laboratories), and DAB peroxidase substrate solution (Vector Laboratories) until desired stain intensity developed. After dehydration in graded ethanol, and vitrification by xylene, sections were mounted. Images were acquired using a 20× objective on a Leica DMi8 inverted microscope. The total number of CHAT-positive cells were counted in each spinal cord section, using three to five sections from each mouse.

## Characterization of retina phenotypes

Eyes were enucleated, and retinae were dissected and fixed in 4% paraformaldehyde in PBS at 4°C overnight. The fixed retinae were cryoprotected in 30% sucrose overnight and embedded in optimal cutting temperature compound. Frozen sections (20 μm thick) were prepared using a Thermo HM525NX cryostat. Retinal sections were blocked in 5% BSA in PBST (PBS with 0.1% Triton X-100), stained with primary antibodies at 4°C overnight, and washed three times with PBST. Sections were then stained using secondary antibodies, including donkey anti-rabbit Alexa Fluor 594, donkey anti-rabbit Alexa Fluor 488, and donkey anti-goat Alexa Fluor 488 (Jackson ImmunoResearch), and were co-stained with DAPI in the dark for overnight at 4°C. Apoptotic cells were evaluated using the TUNEL BrightGreen Apoptosis Detection Kit according to the manufacturer's protocol (Vazyme Biotech). Images were taken using a 20× objective on a Zeiss LSM880 confocal microscope. Images used for comparison between groups were taken side by side using the same confocal settings. ImageJ software was used to quantify relative GFAP immunofluorescent intensity of whole retinal section and thickness of outer nuclear layer (ONL), inner nuclear layer (INL), and inner plexiform layer (IPL) at 1 mm from optic nerve head in retinas. The numbers of IBAl-positive cells, Brn3a-positive cells, and TUNEL-positive cells in each retinal section were quantified

manually. The length of the outer segment of the cone photoreceptors was measured using Nikon NIS Elements AR Offline Station. Three–six eyeballs per group were analyzed for the studies.

### Transmission electron microscopy

Isolated spinal cords were fixed in 2.5% glutaraldehyde, 2% paraformaldehyde in 0.1 M sodium cacodylate/HCl buffer (pH 7.3). After washing with the 0.1 M sodium cacodylate/HCl buffer (pH 7.3), tissues were post-fixed in 1% osmium tetroxide, 1.5% potassium ferricyanide, followed by dehydrating in 25, 50, 70, 95, 100, and 100% alcohol. Finally, the tissues were embedded using the Quetol 651/NSA Embedding Kit (Electron Microscopy Sciences), followed by polymerization in a 60°C oven for 36 h. After polymerization, blocks were sectioned at 70 nm, transferred onto 3 mm copper grids, and stained for 30 min in 1.5% aqueous uranyl acetate. Images were acquired on Morgnani 268 transmission electron microscope (FEI).

### RNA-Seq analysis

RNAs were extracted from spinal cord and cortex regions using TRIzol (Thermo scientific). RNA quality was checked using NanoDrop, gel electrophoresis, and Agilent 4200 TapeStation. cDNA library was then generated using the QuantSeq 3′ mRNA-Seq Library Prep Kit FWD for Illumina (Lexogen). 86 bp single-end sequencing was performed on an Illumina NextSeq500 (Illumina) using services provided by the Cornell Biotech Facilities. Data quality was assessed using FastQC (https://www.bioinformatics.babraham.ac.uk/projects/fastqc/). Reads that passed quality control were aligned to reference genome (Ensembl 98.38) (Zerbino *et al*, 2018) using STAR(Dobin *et al*, 2013). The number of reads in each gene was obtained using HTSeq-count (Anders *et al*, 2015). Differential expression analysis was performed using the R package edgeR (Robinson *et al*, 2010), followed by the limma package with its voom method (Law *et al*, 2014). Genes with FDR control *P*-value ≤ 0.05 and log fold change ≥ 0.5 were identified as differentially expressed genes. Heatmaps were made using the R package gplots. Gene enrichment analysis using cellular component, KEGG pathways, molecular function, and biological process was performed using DAVID 6.8(da Huang *et al*, 2009a,b).

### Statistical analysis

All statistical analyses were performed using GraphPad Prism 8. All data are presented as mean ± SEM. Statistical significance was assessed by unpaired Student's *t*-test (for two group comparison), one-way ANOVA tests with Bonferroni's multiple comparisons (for multiple comparisons), or Fisher's exact test (analysis of contingency tables). *P* values less than or equal to 0.05 were considered statistically significant. *$P < 0.05$; **$P < 0.01$; ***$P < 0.001$; ****$P < 0.0001$.

## Data availability

The data supporting the findings of this study are included in the supplemental material. Additional data are available from the corresponding author on request. No data are deposited in databases.

**Expanded View** for this article is available online.

## Acknowledgements

We would like to thank Xiaochun Wu for technical assistance, Dr. Tony Bretscher's laboratory for assistance with confocal microscope, Dr. Tobias Dörr's laboratory for assistance with Leica DMi8 inverted microscope, Dr. Thomas Cleland's laboratory and Dr. Chris Schaffer and Dr. Nozomi Nishimura's laboratory for assistance with behavioral tests. We would like to acknowledge Dr. Teresa A. Miller from Weill Medical College for help with TEM. This work is supported by NINDS/NIA (R01NS088448 & R01NS095954) and the Bluefield project to cure frontotemporal dementia to F.H.

## Author contributions

TF bred the mice and characterized all the phenotypes except retina with the help from RRS, MU, and IIK. SM characterized retinal phenotypes under the supervision of WX. JMR performed EM analysis. JZ performed bioinformatic analysis for the RNA-Seq data under the supervision of HY. FH supervised the project and wrote the manuscript together with TF. All authors have read and edited the manuscript.

## Conflict of interest

The authors declare that they have no conflict of interest.

## Ethical approval and consent to participate

All applicable international, national, and/or institutional guidelines for the care and use of animals were followed. The work under animal protocol 2017-0056 is approved by the Institutional Animal Care and Use Committee at Cornell University.

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
