## [Review Process File · EMBO Reports]

Loss of TMEM106B and PGRN leads to severe lysosomal abnormalities and neurodegeneration in mice

Tuancheng Feng, Shuyi MAI, Jenn Marie Roscoe, Rory R Sheng, Mohammed Ullah, Junke Zhang, Isabel Katz, Haiyuan Yu, Wenjun Xiong, and Fenghua Hu

DOI: N/A

Corresponding author(s): Fenghua Hu (fh87@cornell.edu)

Review Timeline:

Submission Date:	16th Feb 20
Editorial Decision:	1st Apr 20
Revision Received:	15th May 20
Editorial Decision:	23rd Jun 20
Revision Received:	25th Jun 20
Accepted:	14th Jul 20

Editor: Esther Schnapp

Transaction Report:

Dear Dr. Hu,

Thank you for the submission of your manuscript to EMBO reports. We have now received the enclosed referee reports on it.

As you will see, all referees agree that the findings are novel and interesting. Referee 1 has a number of suggestions for how the study could be further improved, however, upon referee cross-commenting all referees agree that it will be sufficient if these points can be addressed in the manuscript text.

I would therefore like to invite you to revise your manuscript with the understanding that the referee concerns must be fully addressed and their suggestions taken on board. Please address all referee concerns in a complete point-by-point response. Acceptance of the manuscript will depend on a positive outcome of a second round of review.

Revised manuscripts should be submitted within three months of a request for revision; they will otherwise be treated as new submissions.

Regarding data quantification, please specify the number "n" for how many independent experiments were performed, the bars and error bars (e.g. SEM, SD) and the test used to calculate p-values in the respective figure legends. This information must be provided in the figure legends. Please also include scale bars in all microscopy images.

- 1) A data availability section providing access to data deposited in public databases is missing. If you have not deposited any data, please add a sentence to the data availability section that explains that.
- 2) Your manuscript contains statistics and error bars based on $n=2$ or on technical replicates. Please use scatter blots in these cases. No statistics can be calculated if $n=2$.

2) individual production quality figure files as .eps, .tif, .jpg (one file per figure).

See https://wol-prod-cdn.literatuumonline.com/pb-assets/embo-site/EMBOPress_Figure_Guidelines_061115-1561436025777.pdf for more info on how to prepare your figures.

3) We replaced Supplementary Information with Expanded View (EV) Figures and Tables that are collapsible/expandable online. A maximum of 5 EV Figures can be typeset. EV Figures should be cited as 'Figure EV1, Figure EV2' etc... in the text and their respective legends should be included in the main text after the legends of regular figures.

- For the figures that you do NOT wish to display as Expanded View figures, they should be

bundled together with their legends in a single PDF file called *Appendix*, which should start with a short Table of Content. Appendix figures should be referred to in the main text as: "Appendix Figure S1, Appendix Figure S2" etc. See detailed instructions regarding expanded view here: <<https://www.embopress.org/page/journal/14693178/authorguide#expandedview>>

5) a complete author checklist, which you can download from our author guidelines <<https://www.embopress.org/page/journal/14693178/authorguide>>. Please insert information in the checklist that is also reflected in the manuscript. The completed author checklist will also be part of the RPF.

6) Please note that all corresponding authors are required to supply an ORCID ID for their name upon submission of a revised manuscript (<<https://orcid.org/>>). Please find instructions on how to link your ORCID ID to your account in our manuscript tracking system in our Author guidelines <<https://www.embopress.org/page/journal/14693178/authorguide#authorshipguidelines>>

7) Before submitting your revision, primary datasets produced in this study need to be deposited in an appropriate public database (see <https://www.embopress.org/page/journal/14693178/authorguide#datadeposition>). Please remember to provide a reviewer password if the datasets are not yet public. The accession numbers and database should be listed in a formal "Data Availability" section placed after Materials & Method (see also <https://www.embopress.org/page/journal/14693178/authorguide#datadeposition>). Please note that the Data Availability Section is restricted to new primary data that are part of this study. * Note - All links should resolve to a page where the data can be accessed. * If your study has not produced novel datasets, please mention this fact in the Data Availability Section.

8) We would also encourage you to include the source data for figure panels that show essential data. Numerical data should be provided as individual .xls or .csv files (including a tab describing the data). For blots or microscopy, uncropped images should be submitted (using a zip archive if multiple images need to be supplied for one panel). Additional information on source data and instruction on how to label the files are available at <<https://www.embopress.org/page/journal/14693178/authorguide#sourcedata>>.

9) Our journal also encourages inclusion of *data citations in the reference list* to directly cite datasets that were re-used and obtained from public databases. Data citations in the article text are distinct from normal bibliographical citations and should directly link to the database records from which the data can be accessed. In the main text, data citations are formatted as follows: "Data ref: Smith et al, 2001" or "Data ref: NCBI Sequence Read Archive PRJNA342805, 2017". In the Reference list, data citations must be labeled with "[DATASET]". A data reference must provide the database name, accession number/identifiers and a resolvable link to the landing page from which the data can be accessed at the end of the reference. Further instructions are available at

<https://www.embopress.org/page/journal/14693178/authorguide#referencesformat>

I look forward to seeing a revised version of your manuscript when it is ready. Please let me know if you have questions or comments regarding the revision.

Kind regards,
Esther

Referee #1:

This manuscript reports the novel findings that loss of PGRN and TMEM106B leads to exacerbated neurodegenerative phenotypes in mice. Previous GWAS studies have implicated TMEM106B in PGRN expression and in the pathogenesis of frontotemporal lobar degeneration (FTLD). However, the exact mechanism remains unclear. Interestingly, recent results from the Strittmatter Lab (Klein et al., *Neuron* 2017) showed that loss of TMEM106B ameliorates lysosomal phenotypes in *Grn*^{-/-} neurons. In addition, overexpression of TMEM106B in neurons exacerbates lysosomal defects in *Grn*^{-/-} neurons during aging (Zhou et al., *Acta Neuropathol Comm* 2017). Together, these results support that reducing TMEM106B can be protective of neurodegeneration in PGRN deficiency. This study began with a new line of *Tmem106b*^{-/-} mice generated using CRISPR-Cas9, which removed the entire coding region of the mouse *Tmem106b* gene, and was backcrossed into C57Bl6 background for 7 generations to reduce "off target" effects. This new *Tmem106b* line was crossed with *Grn*^{-/-} to generate double KO (dKO) mice. Contrary to the previous results from the Strittmatter Lab, *Tmem106b*^{-/-};*Grn*^{-/-} dKO mice generated in this study developed prominent motor deficits, microgliosis, astrogliosis and loss of neurons in spinal cord, retina and, to a lesser extent, brain. RNA-seq using spinal cord and cortex from 2.7 months wild type, *Tmem106b*^{-/-}, *Grn*^{-/-} and dKO mice showed profound upregulation of inflammatory and lysosomal in the spinal cord of dKO, but much milder transcriptomic changes in the dKO cortex. These results were validated in part via Western blot (WB) and Immunohistochemistry (IHC), which showed abnormal increase in lysosomal enzymes in dKO, primarily affecting microglia and astrocytes. Although similar lysosomal defects in were not detected in spinal cord neurons (subtype?), there was pretty robust accumulation of p62, p-TDP43 and Ubiquitin. In addition, accumulation of lysosome and autophagosome was identified in

the myelinated axons in dKO mice shown by TEM and abnormal increase in LC3-II lipidation via western blots in the spinal cord. Finally, the authors showed that TMEM106B^{-/-} mice have age dependent FTLD-like pathology with increased p62 and p-TDP43 accumulation at 16mo. Overall, this is a very interesting study that provides important insights into the synergistic interactions between TMEM106B and PGRN in maintaining normal lysosomal morphology and function. While the results in this study contradict previous findings by Klein and colleagues, the data are quite compelling and offer new insights into the cooperative role of TMEM106B and PGRN in regulating lysosomal function during brain aging. While I support the publication these findings, the authors need to address several major issues regarding data presentation and interpretation on which cell type(s) are driving neurodegeneration in dKO mice. Given the conflicting data in the literature (some from the authors' own work), it is important the authors provide clarity regarding how their new data will change the way we understand the biology of TMEM106B. The following sections provide specific points on how these weaknesses can be addressed.

Major Points:

1. Figures 2-4: There are several significant issues with the data in these figures. First, NeuN panels in Figure 2A appear to be upside down (ventral side up and dorsal side down), whereas the IBA1 and GFAP panels are rotated 90 degree clockwise (dorsal pointing to the right, and ventral side to left). These are not conventional way to present neuroanatomical/neuropathological results and should be corrected. Second, although the authors show reduced NeuN⁺ cells in "in the middle of spinal cord" in dKO mice, it is unclear exactly which region(s) was counted and how the counts were conducted. The numbers in Figure 2B (y-axis) did not show any unit. The quantification for GFAP and IBA1 was particularly confusing because they did not indicate the cellular density in astrocytes or microglia. Although the graphs in 2B show increases in GFAP and IBA1 intensity in Tmem106b^{-/-} and Grn^{-/-} single mutants, the images in 2A do not support this conclusion. This reviewer believes that quantifications for NeuN⁺, GFAP⁺, IBA1⁺ and ChAT⁺ cells should be conducted using stereology-based counting to provide eliminate any confusion. The same is true for GFAP count in the retina (panel 4F). Third, the western blot results in Figure 3 show reduced PSD95 in spinal cord. This should be supported by histology to show the reduction in synaptic density in spinal cord and which region. Finally, the analyses were conducted in 5 months old mice, which is near disease end stage, but the neuronal loss seems to be much more modest compared to other motor neuron disease models, e.g. SOD1-G93A mice. What do the authors think is the most important neurodegenerative phenotypes that cause mortality in dKO mice?

2. Figure 5: The RNA-seq results in Figure 5 are critical parts of this study and should help understand the phenotypes in the spinal cord of dKO mice. However, the analyses are very superficial and fail to provide mechanistic insights on the mechanism of neurodegeneration in the dKO spinal cord. For instance, the authors should conduct more in-depth bioinformatics analyses (e.g. WGCNA) to indicate which cell type(s) are most responsible for the massively up-regulated genes in dKO spinal cord. Second, panels 5B and 5C are basically the same results, just different emphasis. But, there are also many down-regulated genes in 5B that were not discussed at all. This is a serious shortcoming that should be addressed. I'd encourage the authors to perform Gene Ontology analyses using online resources (e.g. PANTHER, Metascape, KEGG, Cytoscape, etc), and identify specific cellular functions or molecular pathways are selectively affected in the up- and down-regulated gene groups and which cell type(s) are vulnerably affected. Finally, the authors mentioned the transcriptomes of dKO overlap with disease-associated microglia (DAM). However, this statement needs to be examined more rigorously. For instance, it is important to show whether this overlapping is statistically significant? If so, what is the functional implication? If the answer is no, then do microglia in the dKO spinal cord possess its own "disease-specific" signatures?

3. Figures 6-9: To some extent, the results here serve as validations of the transcriptomic work in Figure 5 and the data seem to support the presence of profound lysosomal defects. However, it is not very clear which cell types are affected. For instance, the results in 6E and 8B show that dKO

microglia exhibit profound lysosomal defects. Based on the results in 8A, the authors suggest that dKO astrocytes also have similar lysosomal defects. However, GFAP is not a very good marker to outline the cytoplasm of astrocytes and not all astrocytes are GFAP-positive. Furthermore, the results in Figure 8A, bottom panels for dKO, can also be caused by other cells, such as microglia, that are very close to GFAP+ cells. As such, we do not know what percentage of GFAP+ show increase in lysosomes. Finally, based on the results in Appendix Figure S3, the authors concluded that "lysosomal phenotypes were not observed in neuronal cell bodies". However, their TEM results showed "accumulation of electron dense lysosomes and autophagosomes in myelinated axons". This discrepancy is very confusing and should be clarified by more rigorous examinations in confocal microscopy, TEM or immuno-EM using cell type-specific antibodies (NeuN, IBA1 or GFAP). If the authors re-examine their TEM images, they should be able to identify cell type-specific features.

4. Figures 10-11: The current arrangement for these two figures seems odd and counter-intuitive. If the goal is to show that loss of TMEM106B and PGRN accelerates neurodegeneration and protein aggregate formation in dKO spinal cord, then it is more logical to move Figure 11 to the Appendix and expand it to include data from 5 months old *Tmem106^{-/-}* mice to demonstrate the age dependent changes in TMEM106B *-/-* mice. In addition, the authors should consider including data from 5 and 16 months old *Grn^{-/-}* mice in the same Appendix Figure.

5. Aside from the above issues about data presentation, this manuscript has several significant issues regarding its title, interpretations, and conclusions. First, the current title "Loss of TMEM106B leads to lysosome abnormalities and neurodegeneration in progranulin deficient mice" implies that progranulin deficient mice do not have lysosome abnormalities or neurodegeneration. This is simply incorrect and misleading! Many previous studies (e.g. Lui et al., Cell 2016, Chang et al., JEM 2017, Gotzl et al., Mol Neurodeg 2018, etc) have provided definitive evidence supporting the age-dependent lysosomal abnormalities in *Grn^{-/-}* mice. As such, it is more accurate to revise the title to "Loss of TMEM106B exacerbates lysosome abnormalities and neurodegeneration in progranulin deficient mice", or "Concurrent loss of TMEM106B and Progranulin exacerbates lysosomal abnormalities and neurodegeneration".

6. The authors opened the Discussion by addressing the differences between their results and those from the Strittmatter Lab. However, they never address these new results in the context of their own previous study (Zhou et al., Acta Neuropathol Comm 2017). In light of the new results, is it possible that TMEM106B may have a very tight regulation in its own dosage such that higher or lower TMEM106B may have similar impacts on lysosomal functions?

7. Despite the striking results from histopathology and RNA-seq, the authors seem to be very tentative in their interpretation on the contributions of glial pathologies to neurodegeneration in dKO mice. In Discussion, they stated "At this stage, it is still unclear which cell types are the most affected in *Tmem106b^{-/-}Grn^{-/-}* mice. The neurodegenerative phenotypes could be caused by neuron-intrinsic factors or mis-regulated glial activation". Really? Where is the evidence supporting the presence of "neuron-intrinsic factors" that can promote neurodegeneration in dKO?

8. In Discussion, the authors mentioned "*Tmem106b^{-/-}Grn^{-/-}* mice show a defect in autophagy flow (Fig. 9)". This is over-stating and over-interpreting the results in Figure 9B, which showed increased in lipidated LC3. These results do not address "autophagy flow" or "flux". It simply indicates that the autophagy process is impaired.

9. Finally, the authors should provide a more balanced discussion on the role of progranulin in microglia and lysosomal functions. As mentioned in Point #5, many studies (e.g. Martens et al., JCI 2012, Lui et al., Cell 2016, Chang et al., JEM 2017, Kao et al., NRN 2017, Gotzl et al., Mol Neurodeg 2018, Nguyen et al., PNAS 2018, etc) have provided definitive evidence supporting the age-dependent lysosomal and microglial abnormalities in *Grn^{-/-}* mice and other *Grn* models. It will be important to include these studies in reference citations.

Referee #2:

This is a review of the manuscript by Feng et al entitled "Loss of TMEM106b leads to lysosome abnormalities and neurodegeneration in progranulin deficient mice." It has been firmly established that polymorphisms in TMEM106B dramatically alter the risk of dementia in humans with concurrent GRN mutations. Both genes encode lysosomal proteins, suggesting the possibility of pathogenic convergence. However, direct, compelling evidence of such convergence from cellular or mouse studies has been lacking.

Here, the authors provide compelling evidence that this is indeed the case through a series of experiments in mice lacking GRN, TMEM106B, or both genes. They conclusively show that dual knockdown of GRN and TMEM106B synergistically worsen neurodegenerative phenotypes in the brain, retina, and spinal cord, associated behavioral phenotypes, inflammatory markers, as well as signatures of autophagosomal/lysosomal dysfunction.

These findings are incredibly important for the field, providing the first direct evidence that it is likely that loss of TMEM106B that potentiates GRN insufficiency. These data will open up new lines of research to investigate how such loss mechanistically worsens GRN-related lysosomal dysfunction. I recommend that the manuscript be accepted after minor revisions. I have recommended a number of ways to further improve the manuscript below, which I believe could be accomplished using data that is likely on hand, or simply requires textual revision.

Major comments:

Entire document needs to be reviewed for grammar

One of the longstanding critiques in our field is that only GRN KO mice have significant phenotypes (e.g. GRN het mice are very similar, if not identical, to WT mice). Though not absolutely necessary for publication here, it would be very helpful to the field to include any phenotypic data generated from GRN het/TMEM106B KO or GRN het/TMEM106B het mice. Does full or partial TMEM106B loss "bring out" any relevant pathologic, behavioral, or transcriptional phenotypes of GRN het mice? The authors mention this possibility in the discussion, but if any data are available it would further strengthen the current manuscript.

Minor comments:

Introduction and Abstract:

Clarify what is meant by glial activation

Recommend switching order of paragraphs 3 and 4 of intro

Results:

Why is spinal cord pathology a main focus of results? GRN is not associated with ALS

Why are synapses preserved in CNS but lost in spinal cord.

Clarify: "TUNEL staining revealed increased number [... what...] of cell death in TMEM106b GRN retina"

Clarify: "...data support that ablation of TMEM106b enhances the manifestation of FTLD phenotypes in PGRN deficient background" in a paragraph discussing pathology; pathology¹ phenotype

Comment on studies showing that increased TMEM106b levels also lead to abnormal lysosomes

Focus conclusion paragraph "our studies further underscore the role of lysosome dysfunction in neurodegeneration..." to highlight specific findings and implications (too broad/vague)

Use different combination of colors when depicting inflammation genes in figure 5 (red/green color blindness)

Comment on figure 3C is GRN upregulated in TMEM KO?

Figure 3A GRN gel in SC poor quality

Referee #3:

The authors have generated a new TMEM106b^{-/-}; GRN^{-/-} double knock-out mice model and show that the loss of TMEM106b exacerbates the lysosomal and autophagic dysfunctions in GRN^{-/-} mice leading to motor deficits and neurodegeneration. The paper is clearly written and helps to resolve an area of contradiction in the field.

My comments:

- Introduction: "leads to FTLD like pathology". In fact the observed motor neuron loss, spinal microgliosis and astrogliosis are also reminiscent of ALS, which is not discussed.
- Figure 10 and 11: soluble fraction should be soluble fraction
- The accumulation of ubiquitinated proteins and pTDP-43 is remarkable, as it is the hallmark of FTLD/ALS, which was missing in the GRN^{-/-} mice. This aspect could be discussed more, especially the relation between lysosomal dysfunction, TDP-43 accumulation and PGRN functioning.

We thank the reviewers for their constructive comments on our manuscript. We have revised our manuscript extensively and added new and exciting data.

- 1) We have observed the accumulation of LAMP1 and cathepsin D-positive vacuoles near the distal end of the axon initial segment (AIS) of the motor neurons in the spinal cord sections of TMEM106B deficient mice, consistent with results from a recently published manuscript (Lüningschrör P et al, Cell Rep. 2020 Mar 10;30(10):3506-3519.). Furthermore, we found that this phenotype is further exacerbated in the *Tmem106b*^{-/-} *Grn*^{-/-} mice. These new data have been added to Fig. 1A and Fig. 9A in the revised manuscript.
- 2) We have done gene ontology analyses to show pathways affected in the DKO mice in our RNA-seq experiment. We have also done statistical analyses to show that inflammatory and lysosomal genes are significantly upregulated and genes regulating synapse functions are significantly down-regulated in the *Tmem106b*^{-/-} *Grn*^{-/-} spinal cord samples. These new data are now shown in Fig. EV2.
- 3) We have re-analyzed acquired TEM images and added Fig. 11B to the revised manuscript to show the accumulation of myelin debris and amorphous inclusions in microglia from the *Tmem106b*^{-/-} *Grn*^{-/-} spinal cord sections.

Please see below for detailed responses to the reviewers' comments.

Referee #1:

This manuscript reports the novel findings that loss of PGRN and TMEM106B leads to exacerbated neurodegenerative phenotypes in mice. Previous GWAS studies have implicated TMEM106B in PGRN expression and in the pathogenesis of frontotemporal lobar degeneration (FTLD). However, the exact mechanism remains unclear. Interestingly, recent results from the Strittmatter Lab (Klein et al., *Neuron* 2017) showed that loss of TMEM106B ameliorates lysosomal phenotypes in *Grn*^{-/-} neurons. In addition, overexpression of TMEM106B in neurons exacerbates lysosomal defects in *Grn*^{-/-} neurons during aging (Zhou et al., *Acta Neuropathol Comm* 2017). Together, these results support that reducing TMEM106B can be protective of neurodegeneration in PGRN deficiency.

This study began with a new line of *Tmem106b*^{-/-} mice generated using CRISPR-Cas9, which removed the entire coding region of the mouse *Tmem106b* gene, and was backcrossed into C57Bl6 background for 7 generations to reduce "off target" effects. This new *Tmem106b* line was crossed with *Grn*^{-/-} to generate double KO (dKO) mice. Contrary to the previous results from the Strittmatter Lab, *Tmem106b*^{-/-}; *Grn*^{-/-} dKO mice generated in this study developed prominent motor deficits, microgliosis, astrogliosis and loss of neurons in spinal cord, retina and, to a lesser extent, brain. RNA-seq using spinal cord and cortex from 2.7 months wild type, *Tmem106b*^{-/-}, *Grn*^{-/-} and dKO mice showed profound upregulation of inflammatory and lysosomal in the spinal cord of dKO, but much milder transcriptomic changes in the dKO cortex. These results were validated in part via Western blot (WB) and Immunohistochemistry (IHC), which showed abnormal increase in lysosomal enzymes in dKO, primarily affecting microglia and astrocytes. Although similar lysosomal defects in were not detected in spinal cord neurons (subtype?), there was pretty robust accumulation of p62, p-TDP43 and Ubiquitin. In addition, accumulation of lysosome and autophagosome was identified in the myelinated axons in dKO mice shown by TEM and abnormal increase in LC3-II lipidation via western blots in the spinal cord. Finally, the authors showed that TMEM106B^{-/-} mice have age dependent FTLD-like pathology with increased p62 and p-TDP43 accumulation at 16mo.

Overall, this is a very interesting study that provides important insights into the synergistic interactions between TMEM106B and PGRN in maintaining normal lysosomal morphology and function. While the results in this study contradict previous findings by Klein and colleagues, the data are quite compelling and offer new insights into the cooperative role of TMEM106B and PGRN in regulating lysosomal function during brain aging. While I support the publication these findings, the authors need to address several major issues regarding data presentation and interpretation on which cell type(s) are driving neurodegeneration in dKO mice. Given the conflicting data in the literature (some from the authors' own work), it is important the authors provide clarity regarding how their new data will change the way we understand the biology of TMEM106B.

Reply: We thank the reviewer for the careful and thorough review of our manuscript.

The following sections provide specific points on how these weaknesses can be addressed.

Major Points:

1. Figures 2-4: There are several significant issues with the data in these figures. First, NeuN panels in Figure 2A appear to be upside down (ventral side up and dorsal side down), whereas the IBA1 and GFAP panels are rotated 90 degree clockwise (dorsal pointing to the right, and ventral side to left). These are not conventional way to present neuroanatomical/neuropathological results and should be corrected. Second, although the authors show reduced NeuN+ cells in "in the middle of spinal cord" in dKO mice, it is unclear exactly which region(s) was counted and how the counts were conducted. The numbers in Figure 2B (y-axis) did not show any unit. The quantification for GFAP and IBA1 was particularly confusing because they did not indicate the cellular density in astrocytes or microglia. Although the graphs in 2B show increases in GFAP and IBA1 intensity in *Tmem106b*^{-/-} and *Grn*^{-/-} single mutants, the images in 2A do not support this conclusion. This reviewer believes that quantifications for NeuN+, GFAP+, IBA1+ and ChAT+ cells should be conducted using stereology-based counting to provide eliminate any confusion. The same is true for GFAP count in the retina (panel 4F). Third, the western blot results in Figure 3 show reduced PSD95 in spinal cord. This should be supported by histology to show the reduction in synaptic density in spinal cord and which region. Finally, the analyses were conducted in 5 months old mice, which is near disease end stage, but the neuronal loss seems to be much more modest compared to other motor neuron disease models, e.g. *SOD1-G93A* mice. What do the authors think is the most important neurodegenerative phenotypes that cause mortality in dKO mice?

Reply: (1) We have edited the images according to the reviewer's suggestions. (2) Quantification of the number of NeuN+ cells was done using the entire spinal cord section, not just the middle region. Regarding the Iba1 and GFAP quantifications, the Y-axis is the total intensity of all of the Iba1/GFAP-positive signals in the entire spinal cord section. GFAP intensity in the retina was quantified similarly. Shown in Fig. 4B is the total intensity of GFAP in one retinal section. We have changed the labels in the figures to avoid confusion. (3) The reduction of the intensity of PSD95 is consistent with the reduction of NeuN-positive neurons in the spinal cord. Future work will be needed to determine which cell types are mostly affected in the spinal cord and in the brain. (4) The mortality is likely caused by neuronal cell death. Based on our analysis so far, the loss of motor neurons in the spinal cord is likely to contribute to the mortality of the dKO mice. But the loss is modest compared to the *SOD1-G93A* mice, as pointed out by the reviewer. It is possible that lysosome trafficking defects due to the loss of TMEM106B further leads to neuronal dysfunction without causing neuronal death. We plan to thoroughly examine the types of neurons affected using specific markers in the future.

2. Figure 5: The RNA-seq results in Figure 5 are critical parts of this study and should help understand the phenotypes in the spinal cord of dKO mice. However, the analyses are very superficial and fail to provide mechanistic insights on the mechanism of neurodegeneration in the dKO spinal cord. For instance, the authors should conduct more in-depth bioinformatics analyses (e.g. WGCNA) to indicate which cell type(s) are most responsible for the massively up-regulated genes in dKO spinal cord. Second, panels 5B and 5C are basically the same results, just different emphasis. But, there are also many down-regulated genes in 5B that were not discussed at all. This is a serious shortcoming that should be addressed. I'd encourage the authors to perform Gene Ontology analyses using online resources (e.g. PANTHER, Metascape, KEGG, Cytoscape, etc), and identify specific cellular functions or molecular pathways are selectively affected in the up- and down-regulated gene groups and which cell type(s) are vulnerably affected. Finally, the authors mentioned the transcriptomes of dKO overlap with disease-associated microglia (DAM). However, this statement needs to be examined more rigorously. For instance, it is important to show whether this overlapping is statistically significant? If so, what is the functional implication? If the answer is no, then do microglia in the dKO spinal cord possess its own "disease-specific" signatures?

Reply: We totally agree with the reviewer. We have re-analyzed the data according to the reviewer's suggestions and the new analyses are shown in Fig. EV2. In Fig. 6, panel 5B is the overall heatmap which contains both up- and down-regulated differentially expressed genes (DEGs) in the spinal cord, whereas panel 5C highlights up-regulated lysosome and disease associated microglia (DAM) genes in the DKO, since our pathway analysis has identified the lysosomal and inflammatory pathways as the two pathways mostly affected in the DKO spinal cord (Fig. EV2A). We've also performed gene enrichment analysis of the downregulated genes and found that genes functioning in the synapse are among the most downregulated group (Fig. EV2B), which is consistent with the loss of PSD95 and NeuN signals in the Western blot analysis (Figs. 3 and 4A). Additionally, we've statistically analyzed DAM genes and found that they are significantly upregulated in the DKO (Figs. 6C and EV2D). Our RNA seq analysis would suggest that both neurons and microglia are vulnerably affected in the DKO.

3. Figures 6-9: To some extent, the results here serve as validations of the transcriptomic work in Figure 5 and the data seem to support the presence of profound lysosomal defects. However, it is not very clear which cell types are affected. For instance, the results in 6E and 8B show that dKO microglia exhibit profound lysosomal defects. Based on the results in 8A, the authors suggest that dKO astrocytes also have similar lysosomal defects. However, GFAP is not a very good marker to outline the cytoplasm of astrocytes and not all astrocytes are GFAP-positive. Furthermore, the results in Figure 8A, bottom panels for dKO, can also be caused by other cells, such as microglia, that are very close to GFAP+ cells. As such, we do not know what percentage of GFAP+ show increase in lysosomes. Finally, based on the results in Appendix Figure S3, the authors concluded that "lysosomal phenotypes were not observed in neuronal cell bodies". However, their TEM results showed "accumulation of electron dense lysosomes and autophagosomes in myelinated axons". This discrepancy is very confusing and should be clarified by more rigorous examinations in confocal microscopy, TEM or immuno-EM using cell type-specific antibodies (NeuN, IBA1 or GFAP). If the authors re-examine their TEM images, they should be able to identify cell type-specific features.

Reply: Our RNA seq analyses indicate that both neurons (Fig. EV2B) and microglia (Fig. EV2A) are affected in the DKO. Regarding neuronal pathology, we have observed the accumulation of LAMP1 and cathepsin D-positive vacuoles near the distal end of the AIS of motor neurons in the spinal cord sections of TMEM106B-deficient mice and DKO mice. This is consistent with a recent Cell Report paper (Lüningschrör P et al, Cell Rep. 2020 Mar 10;30(10):3506-3519.). This new data has been added as Fig. 1A and Fig. 9A. The presence of autophagosomes and lysosomes in the myelinated axons would suggest additional trafficking defects in the axons besides those in the AIS region.

Regarding microglial pathology, we have analyzed more images from the TEM and have observed the accumulation of myelin debris and amorphous inclusions in microglia in the DKO spinal cord section samples. We have added that data in Fig. 11B.

Regarding astrocyte phenotypes, glial fibrillary acidic protein (GFAP) is widely considered a specific marker for astrocytes in the central nervous system. In DKO mice, all of the GFAP-positive cells show lysosome abnormalities. Although we cannot rule out the slight possibility of overlapping signals from neighboring cells, all of the immunofluorescent images were taken with the spinning disk confocal microscope. We feel confident that the majority of the signals are from the indicated cell types.

4. Figures 10-11: The current arrangement for these two figures seems odd and counter-intuitive. If the goal is to show that loss of TMEM106B and PGRN accelerates neurodegeneration and protein aggregate formation in dKO spinal cord, then it is more logical to move Figure 11 to the Appendix and expand it to include data from 5 months old Tmem106^{-/-} mice to demonstrate the age dependent changes in TMEM106B ^{-/-} mice. In addition, the authors should consider including data from 5 and 16 months old Grn^{-/-} mice in the same Appendix Figure.

Reply: We agree with the reviewer that the order of these two figures are a little odd. Since we have also observed the lysosome trafficking defects in the AIS region in TMEM106B-deficient mice, we've decided to move this data (accumulation of ubiquitinated proteins, p62 and phosphor-TDP-43 in the 16-month-old TMEM106B deficient mice) to Figure 1 (Fig. 1B-1E) to be with the AIS data to show the phenotypes of TMEM106B-deficient mice before describing our characterization of the DKO mice. Since the 5-month-old TMEM106B-deficient mice do not show strong phenotypes regarding ubiquitin and TDP-43 pathology (Fig. 12), we decided not to show this in Fig. 1 by itself. We also agree with the reviewer that it would be nice to have the data from 16 month-old Grn^{-/-} mice. Unfortunately, we do not have the samples available for analysis right now.

5. Aside from the above issues about data presentation, this manuscript has several significant issues regarding its title, interpretations, and conclusions. First, the current title "Loss of TMEM106B leads to lysosome abnormalities and neurodegeneration in progranulin deficient mice" implies that progranulin deficient mice do not have lysosome abnormalities or neurodegeneration. This is simply incorrect and misleading! Many previous studies (e.g. Lui et al., Cell 2016, Chang et al., JEM 2017, Gotzl et al., Mol Neurodegen 2018, etc) have provided definitive evidence supporting the age-dependent lysosomal abnormalities in Grn^{-/-} mice. As such, it is more accurate to revise the title to "Loss of TMEM106B exacerbates lysosome abnormalities and neurodegeneration in progranulin deficient mice", or "Concurrent loss of TMEM106B and Progranulin exacerbates lysosomal abnormalities and neurodegeneration".

Reply: Thank you so much for the suggestion. We are fully aware that there are lysosome abnormalities in PGRN-deficient mice. The original title of our manuscript was "Loss of TMEM106B exacerbates lysosome abnormalities and neurodegeneration in progranulin-deficient mice ". We had to change that due to the number of characters allowed by the journal. We've now changed the title to "Loss of TMEM106B and PGRN leads to severe lysosomal abnormalities and neurodegeneration in mice".

6. The authors opened the Discussion by addressing the differences between their results and those from the Strittmatter Lab. However, they never address these new results in the context of their own previous study (Zhou et al., Acta Neuropathol Comm 2017). In light of the new results, is it possible that TMEM106B may have a very tight regulation in its own dosage such that higher or lower TMEM106B may have similar impacts on lysosomal functions?

Reply: Yes, we think that the levels of TMEM106B need to be tightly regulated. Either too much or too little might interfere with lysosomal function. We have added that to the discussion. Thank you very much for the suggestion.

7. Despite the striking results from histopathology and RNA-seq, the authors seem to be very tentative in their interpretation on the contributions of glial pathologies to neurodegeneration in dKO mice. In Discussion, they stated "At this stage, it is still unclear which cell types are the most affected in Tmem106b^{-/-}Grn^{-/-} mice. The neurodegenerative phenotypes could be caused by neuron-intrinsic factors or mis-regulated glial activation". Really? Where is the evidence supporting the presence of "neuron-intrinsic factors" that can promote neurodegeneration in dKO?

Reply: Both PGRN and TMEM106B are expressed in neurons and are important for proper lysosomal function in neurons. Our new data have shown a critical role of TMEM106B in lysosome trafficking at the distal end of axon initial segments of motor neurons (Fig. 1A). Ablation of both PGRN and TMEM106B exacerbates the lysosome trafficking defects (Fig. 9). Our RNA seq analysis, Western blot analysis and immunostaining results all suggest that both neurons and glia are affected by the loss of PGRN and TMEM106B. It is likely that both neuronal and glial dysfunction/mis-regulation lead to neurodegenerative phenotypes. We've edited our discussion accordingly.

8. In Discussion, the authors mentioned "Tmem106b^{-/-}Grn^{-/-} mice show a defect in autophagy flow (Fig. 9)". This is over-stating and over-interpreting the results in Figure 9B, which showed increased in lipidated LC3. These results do not address "autophagy flow" or "flux". It simply indicates that the autophagy process is impaired.

Reply: We completely agree with the reviewer and have revised the text accordingly.

9. Finally, the authors should provide a more balanced discussion on the role of progranulin in microglia and lysosomal functions. As mentioned in Point #5, many studies (e.g. Martens et al., JCI 2012, Lui et al., Cell 2016, Chang et al., JEM 2017, Kao et al., NRN 2017, Gotz et al., Mol Neurodegen 2018, Nguyen et al., PNAS 2018, etc) have provided definitive evidence supporting the age-dependent lysosomal and microglial abnormalities in Grn^{-/-} mice and other Grn models. It will be important to include these

studies in reference citations.

Reply: Thanks for the suggestion. We've added these references in the discussion.

Referee #2:

This is a review of the manuscript by Feng et al entitled "Loss of TMEM106b leads to lysosome abnormalities and neurodegeneration in progranulin deficient mice." It has been firmly established that polymorphisms in TMEM106B dramatically alter the risk of dementia in humans with concurrent GRN mutations. Both genes encode lysosomal proteins, suggesting the possibility of pathogenic convergence. However, direct, compelling evidence of such convergence from cellular or mouse studies has been lacking.

Here, the authors provide compelling evidence that this is indeed the case through a series of experiments in mice lacking GRN, TMEM106B, or both genes. They conclusively show that dual knockdown of GRN and TMEM106B synergistically worsen neurodegenerative phenotypes in the brain, retina, and spinal cord, associated behavioral phenotypes, inflammatory markers, as well as signatures of autophagosomal/lysosomal dysfunction.

These findings are incredibly important for the field, providing the first direct evidence that it is likely that loss of TMEM106B that potentiates GRN insufficiency. These data will open up new lines of research to investigate how such loss mechanistically worsens GRN-related lysosomal dysfunction. I recommend that the manuscript be accepted after minor revisions. I have recommended a number of ways to further improve the manuscript below, which I believe could be accomplished using data that is likely on hand, or simply requires textual revision.

Reply: Thank you very much for the positive and insightful review.

Major comments:

Entire document needs to be reviewed for grammar

Reply: Thank you for the comment. We've tried our best to correct grammar errors in the revision. Sorry about the errors.

One of the longstanding critiques in our field is that only GRN KO mice have significant phenotypes (e.g. GRN het mice are very similar, if not identical, to WT mice). Though not absolutely necessary for publication here, it would be very helpful to the field to include any phenotypic data generated from GRN het/TMEM106B KO or GRN het/TMEM106B het mice. Does full or partial TMEM106B loss "bring out" any relevant pathologic, behavioral, or transcriptional phenotypes of GRN het mice? The authors mention this possibility in the discussion, but if any data are available it would further strengthen the current manuscript.

Reply: Thank you for agreeing with us on the importance of characterizing phenotypes in these mice. Unfortunately, we have not gotten a chance to fully characterize these mice. However, we have not noticed any severe behavioral deficits in these mice even at 16 months of age. We have added that in the text.

Minor comments:

Introduction and Abstract:

Clarify what is meant by glial activation

Reply: Glial activation means activation of microglia and astrocytes. We've made the changes in the text.

Recommend switching order of paragraphs 3 and 4 of intro

Reply: Thank you. We have modified the text accordingly.

Results:

Why is spinal cord pathology a main focus of results? GRN is not associated with ALS

Why are synapses preserved in CNS but lost in spinal cord.

Reply: Thank you very much for pointing this out. We are puzzled by this result as well. However, we've found that TMEM106B deficiency specifically affects lysosome trafficking in the AIS region of motor neurons (Fig. 1A), consistent with a recent report (Lüningschrör P et al, Cell Rep. 2020 Mar 10;30(10):3506-3519.). So, some of the phenotypes observed in the double knockout mice could be due to lysosome trafficking defects in the motor neurons caused by the loss of TMEM106B. Loss of synapses could be secondary to lysosome trafficking defects and neuronal cell death, which is likely to be more affected in motor neurons due to the loss of TMEM106B as explained above.

Clarify: "TUNEL staining revealed increased number [... what...] of cell death in TMEM106b GRN retina"

Reply: We have changed this to "TUNEL staining revealed increased apoptosis in TMEM106b GRN retina" since we are not sure which cell types have undergone apoptosis.

Clarify: "...data support that ablation of TMEM106b enhances the manifestation of FTLN phenotypes in PGRN deficient background" in a paragraph discussing pathology; pathology¹ phenotype

Reply: Sorry for the confusion. We were referring to molecular changes seen in FTLN patients, such as accumulation of ubiquitinated proteins, p62 and phospho-TDP-43.

Comment on studies showing that increased TMEM106b levels also lead to abnormal lysosomes

Reply: Thanks. We have added that in the discussion.

Focus conclusion paragraph "our studies further underscore the role of lysosome dysfunction in neurodegeneration..." to highlight specific findings and implications (too broad/vague)

Reply: Thanks for the suggestions. We've modified the text.

Use different combination of colors when depicting inflammation genes in figure 5 (red/green color blindness)

Reply: Thanks for the suggestions. We've edited the figure accordingly.

Comment on figure 3C is GRN upregulated in TMEM KO?

Reply: The levels of PGRN and granulin peptides do not appear to be upregulated in TMEM106B-deficient mice. We have added that in the discussion.

Figure 3A GRN gel in SC poor quality

Reply: Thanks, we've re-run the Western blot and replaced the gel in Fig. 4A and 8E.

Referee #3:

The authors have generated a new TMEM106b^{-/-}; GRN^{-/-} double knock-out mice model and show that the loss of TMEM106b exacerbates the lysosomal and autophagic dysfunctions in GRN^{-/-} mice leading to motor deficits and neurodegeneration. The paper is clearly written and helps to resolve an area of contradiction in the field.

Reply: Thank you for your positive comments.

My comments:

- Introduction: "leads to FTLD like pathology". In fact the observed motor neuron loss, spinal microgliosis and astrogliosis are also reminiscent of ALS, which is not discussed.

Reply: Thanks for pointing this out. We have added another paragraph in the discussion.

- Figure 10 and 11: soluble fraction should be soluble fraction

Reply: Thanks! We've corrected that in the figures.

- The accumulation of ubiquitinated proteins and pTDP-43 is remarkable, as it is the hallmark of FTLD/ALS, which was missing in the GRN^{-/-} mice. This aspect could be discussed more, especially the relation between lysosomal dysfunction, TDP-43 accumulation and PGRN functioning.

Reply: Thanks. We have added that in the discussion.

Dear Dr. Hu,

Thank you for the submission of your revised manuscript. We have now received the enclosed report from referee 1 who was asked to assess it. Referee 1 still has one more suggestion that I would like you to address before we can proceed with the official acceptance of your manuscript.

A few other changes will also be required:

Please add up to 5 keywords to the manuscript file.

The Data Availability Section (DAS) should list data that are deposited in public databases, e.g. if your RNA-seq data are deposited somewhere, please add the exact link to the DAS. If no data are deposited in databases, please add this information to the DAS.

Please remove the list of "Appendix materials" from the manuscript file.

Please change the reference style to our new Harvard style. A link to the style can be found in our guide to authors online.

Please remove DATA not shown on page 6 as per journal policy.

Fig 5I+J are called out after Fig 7, please correct.

Please upload the source data as one file per figure.

I attach to this email a related ms file with comments by our data editors. Please address all comments in the final manuscript file.

EMBO press papers are accompanied online by A) a short (1-2 sentences) summary of the findings and their significance, B) 2-3 bullet points highlighting key results and C) a synopsis image that is 550x200-400 pixels large (the height is variable). You can either show a model or key data in the synopsis image. Please note that text needs to be readable at the final size. Please send us this information along with the revised manuscript.

I look forward to seeing a final version of your manuscript as soon as possible. Please let me know if you have any questions or comments.

Referee #1:

The revised manuscript by Feng and colleagues addressed my previous comments by revising figure panels and texts. Most of these revisions appropriately addressed my previous critiques.

In addition, the authors rearranged/revised Figure 1 by adding new Figure 1A that showed loss of TMEM106B affected axon initial segment (AIS). On page 5, first paragraph, they stated "Immunostaining with the axon initial segment (AIS) marker neurofascin revealed that these vacuoles often accumulate at the distal end of the AIS in motor neurons (Fig. 1A), consistent with the results from a recently published study [58]. This suggests a critical role of TMEM106B in mediating lysosome trafficking across AIS specifically in motor neurons."

However, there several issues with the new Figure 1A. First, the immunostaining for neurofascin in Figure 1A lacks the characteristic feature commonly seen in AIS (e.g. Figure 3C in Zhang et al., J Neurosci 2015, 35(5):2246-2254 or Figure 9A in the revised manuscript). The putative phenotype in Figure 1A is also very different from those reported by Luningschror et al, where the authors used Ankryin G and α IV-spectrin to nicely highlight the AIS defects in Tmem106b^{-/-} neurons. Second, the immunostainings for LAMP1 and Cath D in Figure 1A look nearly identical between WT and Tmem106B^{-/-} neurons. The lack of any quantitative analysis of lysosomes in the cytoplasm, AIS or axons makes it very difficult to appreciate the phenotype in AIS in Tmem106^{-/-} neurons.

Given these issues, if the authors think the results in Figure 1A are absolutely essential for the entire story, they should replace the images in Figure 1A with new ones and provide quantification of the lysosomal defects to support their conclusion. Alternatively, they can just remove Figure 1A and the associated texts if these results are not essential to the whole story.

Referee #1:

The revised manuscript by Feng and colleagues addressed my previous comments by revising figure panels and texts. Most of these revisions appropriately addressed my previous critiques.

In addition, the authors rearranged/revised Figure 1 by adding new Figure 1A that showed loss of TMEM106B affected axon initial segment (AIS). On page 5, first paragraph, they stated "Immunostaining with the axon initial segment (AIS) marker neurofascin revealed that these vacuoles often accumulate at the distal end of the AIS in motor neurons (Fig. 1A), consistent with the results from a recently published study [58]. This suggests a critical role of TMEM106B in mediating lysosome trafficking across AIS specifically in motor neurons."

However, there several issues with the new Figure 1A. First, the immunostaining for neurofascin in Figure 1A lacks the characteristic feature commonly seen in AIS (e.g. Figure 3C in Zhang et al., J Neurosci 2015, 35(5):2246-2254 or Figure 9A in the revised manuscript). The putative phenotype in Figure 1A is also very different from those reported by Luningschror et al, where the authors used Ankryin G and β IV-spectrin to nicely highlight the AIS defects in Tmem106b^{-/-} neurons. Second, the immunostainings for LAMP1 and Cath D in Figure 1A look nearly identical between WT and Tmem106B^{-/-} neurons. The lack of any quantitative analysis of lysosomes in the cytoplasm, AIS or axons makes it very difficult to appreciate the phenotype in AIS in Tmem106^{-/-} neurons.

Given these issues, if the authors think the results in Figure 1A are absolutely essential for the entire story, they should replace the images in Figure 1A with new ones and provide quantification of the lysosomal defects to support their conclusion. Alternatively, they can just remove Figure 1A and the associated texts if these results are not essential to the whole story.

Reply: We thank the reviewer for his/her critical review of our manuscript. We have re-examined the AIS phenotypes in our Tmem106b^{-/-} spinal cord sections and compared our staining with other reported AIS images in literature. We feel confident about the specificity of our neurofascin staining although it does have some background signals. The enlarged LAMP1 and CathD positive vacuoles at the distal end of AIS are only observed in Tmem106b^{-/-} neurons, which we have quantified in Fig. 9B. To better describe this phenotype according to the reviewer's request, we have re-plotted this against WT and shown this data in Fig. 1B. We have also replaced the image in Fig. 1A with a more representative one. We agree with the reviewer that it will be nice to quantify other aspects of lysosome changes in Tmem106b^{-/-} as well. However, this is extremely hard to do with mouse tissue sections since it's almost impossible to get images for the entire cell body and axons, if there are subtle changes. Since the accumulation of LAMP1/CathD positive vacuoles are the most obvious phenotype we have observed, we've focused on the quantification of this phenotype in tmem106b^{-/-} and tmem106b^{-/-} grn^{-/-} mice for this manuscript.

We do feel it's important to describe this phenotype in Tmem106b^{-/-} first before touching on this in the DKO mice.

Dr. Fenghua Hu
Cornell University
Weill Institute for Cell and Molecular Biology and Department of Molecular Biology and Genetics
345 Weill Hall, Cornell University
Ithaca, NY 14853
United States

Dear Fenghua,

I am very pleased to accept your manuscript for publication in the next available issue of EMBO reports. Thank you for your contribution to our journal.

At the end of this email I include important information about how to proceed. Please ensure that you take the time to read the information and complete and return the necessary forms to allow us to publish your manuscript as quickly as possible.

As part of the EMBO publication's Transparent Editorial Process, EMBO reports publishes online a Review Process File to accompany accepted manuscripts. As you are aware, this File will be published in conjunction with your paper and will include the referee reports, your point-by-point response and all pertinent correspondence relating to the manuscript.

If you do NOT want this File to be published, please inform the editorial office within 2 days, if you have not done so already, otherwise the File will be published by default [contact: emboreports@embo.org]. If you do opt out, the Review Process File link will point to the following statement: "No Review Process File is available with this article, as the authors have chosen not to make the review process public in this case."

Should you be planning a Press Release on your article, please get in contact with emboreports@wiley.com as early as possible, in order to coordinate publication and release dates.

Thank you again for your contribution to EMBO reports and congratulations on a successful publication. Please consider us again in the future for your most exciting work.

THINGS TO DO NOW:

You will receive proofs by e-mail approximately 2-3 weeks after all relevant files have been sent to

our Production Office; you should return your corrections within 2 days of receiving the proofs.

Please inform us if there is likely to be any difficulty in reaching you at the above address at that time. Failure to meet our deadlines may result in a delay of publication, or publication without your corrections.

All further communications concerning your paper should quote reference number EMBOR-2020-50219V3 and be addressed to emboreports@wiley.com.

Should you be planning a Press Release on your article, please get in contact with emboreports@wiley.com as early as possible, in order to coordinate publication and release dates.

Corresponding Author Name: Fenghua Hu

Journal Submitted to: EMBO reports

Manuscript Number: EMBOR-2020-50219V1